# AUGMENTATIONS IN OFFLINE REINFORCEMENT LEARNING FOR ACTIVE POSITIONING

## ABSTRACT

We propose a method for data augmentation in offline reinforcement learning applied to active positioning problems. The approach enables the training of off-policy models from a limited number of trajectories generated by a suboptimal logging policy. Our method is a trajectory-based augmentation technique that exploits task structure and quantifies the effect of admissible perturbations on the data using the geometric interplay of properties of the reward, the value function, and the logging policy. Moreover, we show that by training an off-policy model with our augmentation while collecting data, the suboptimal logging policy can be supported during collection, leading to higher data quality and improved offline reinforcement learning performance. We provide theoretical justification for these strategies and validate them empirically across positioning tasks of varying dimensionality and under partial observability.

## 1 INTRODUCTION

In active positioning, an end-effector must place an object precisely at a desired pose. Such problems occur in high-precision manufacturing, e.g., in camera Bräuniger et al. (2014) or telescope assembly Upton et al. (2006), in alignments of laser optics Rakhmatulin et al. (2024), as well as in many robotic manipulation tasks Plappert et al. (2018). In optical systems, this involves iterative adjustment of components, such as lenses or mirrors, to maximize alignment quality from image-based signals. These tasks are naturally modeled as contextual partially observed Markov decision problems (POMDPs) and demand generalization over the context Burkhardt et al. (2025). While reinforcement learning (RL) has advanced algorithmically, online training is costly: explorations in high-dimensional observation and continuous action spaces are inefficient, cause long downtimes, and human interaction is often required between episodes, for instance to insert new objects. At the same time, precise but inefficient expert routines can provide data, making offline RL a promising alternative, which sidesteps online interactions by training from pre-collected datasets(Levine et al., 2020). Although offline RL promises is to learn policies better than the logging policy from static datasets, without online interactions, it suffers from distributional shifts and inappropriate datasets, leading to suboptimal policies. To cope with distribution shift, contemporary offline RL methods regularize policies toward the behavior distribution or warm-start from the logging policy before cautiously improving it (see Section 1.2 for an overview).

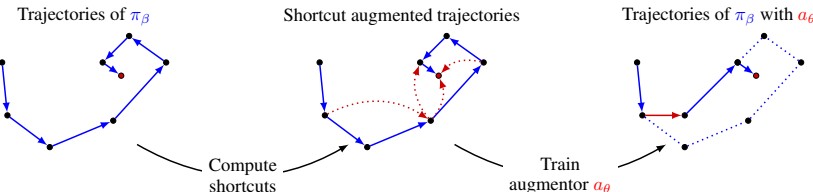

Figure 1: Overview of LIFT.

Despite these algorithmic advances, it remains unclear how the data-generating logging policy limits what an offline learner can achieve. Prior evidence already shows that dataset selection can dominate algorithmic differences (Schweighofer et al., 2022; Fu et al., 2021; Yarats et al., 2022); actionable

guidance for improving the data itself, however, remains scarce. Moreover, when probing effects of logging policies, prior work typically uses different categories of expertness where often the data of highest expertness is produced by an RL agent trained online (Fu et al., 2021). Although this schema is convenient, it could introduce a methodology bias: the generated trajectory inherits the exploration style and failure modes of the training algorithm, not those of deterministic, production-grade expert routines common in practice. Mixing datasets of different quality not only degenerates performance, it can also practically be infeasible to hand-off between policies. For instance, expert systems typically are deterministic, tightly scripted routines with internal states where decisions can depend on the entire trajectory of measurements. Inserting actions in-between can invalidate the routine's assumptions and it only be able to resume reliably if it restarts from the new state.

In this work, we design data-efficient RL methods for active positioning tasks that can effectively learn from *inefficient* expert policies that can precisely place objects but need many steps to do so. Our key idea is to augment the logging policy sparingly with actions proposed by an off-policy learner trained parallel to the data collection. The off-policy learner is trained in a way to explore shortcuts in the experts trajectories to make hand-offs seamless and effective.

## 1.1 CONTRIBUTIONS

We introduce *LIFT*, short for logging improvement via fine-tuned trajectories, a framework that enhances punctual data collection for offline RL. Specifically, we propose a novel augmentation scheme (Section 4) that keeps the logging policy in control while enabling optimistic probing by an augmentor trained as data is collected. The augmentor's goal is to skip redundant and unnecessary sub-trajectories during collection and to smooth hand-offs between itself and the logging policy. A key challenge is to train the augmentor with very limited data so that such *shortcuts* can be identified punctually. Our idea is to train the augmentor on synthetic trajectories obtained from real data that exhibit such shortcuts. We prove in Section 3 under which conditions such shortcuts can be reliably identified in logged data, and we devise an algorithm to extract them from this data (Algorithm 1). Finally, Section 5 presents a systematic study that underlines the strength and generality of our approach by analyzing the effect of the logging policy, transition behavior, dimensionality, and informativeness of observations on policy performance across a diverse class of active positioning tasks. We implemented the shortcut augmentation in d3rlpy Seno & Imai (2022), following its transition picker protocol, which allows our static augmentation method to be integrated into any RL algorithm implemented in d3rlpy by adding a single line of code.[1]

## 1.2 RELATED WORK

A central challenge in offline RL is overestimating values for out-of-distribution actions. Methods address this either by constraining the learned policy toward the logging distribution or by learning pessimistic value functions. Representative approaches include behavior regularization via BC losses or divergence penalties (Fujimoto et al., 2019; Fujimoto & Gu, 2021; Tarasov et al., 2023), pessimistic critics (Kumar et al., 2020), or expectile-based policy extraction (Kostrikov et al., 2022). Methods depending on regularizations are sensitive to hyperparameters and they often limit the policy to stay close to the behavior, for instance due to safety constraints, which can be detrimental if the behavior is highly suboptimal. Moreover, several studies note that algorithm performance is highly sensitive to dataset composition (Fu et al., 2021; Hong et al., 2023), that is, mixing suboptimal trajectories with expert data. Prior work has studied intensively the importance of high-coverage Yarats et al. (2022); Wagenmaker et al. (2025) and expertness of datasets Kumar et al. (2022); Corrado et al. (2024) for offline RL. This has been underpinned by the investigations in Schweighofer et al. (2022), where scores are designed that measure exploitation and exploration capabilities of datasets and how these affect algorithmic performance of offline RL methods. Increasing the dataset diversity via data augmentations is another line of work to mitigate narrow data distributions. In Andrychowicz et al. (2017), an augmentation scheme for sparse reward in robotic manipulation tasks is proposed that re-labels goals and states in logged trajectories to create additional successful transitions. Augmentations for problems with image observations have been studied extensively in the literature, were it was shown that rather simple image augmentations Laskin et al. (2020); Sinha et al. (2022), like

---

[1]The implementations are included in the supplemental material of this submission and will be made available on GitHub upon acceptance.

random or utilizing causal techniques Pitis et al. (2020) can significantly improve sample efficiency. Recently, diffusion-based techniques have been proposed that generate synthetic trajectories in order to make offline RL more robust Li et al. (2024); Lee et al. (2024); Lu et al. (2023). Different than purely *offline* augmentations generating synthetic data from static datasets and more relevant for our work are hybrid schemes actively enhance the data collection process itself. The easiest hybrid scheme is to warm-start online RL from an offline-trained policy, then continue by adding newly collected online. Prior work shows that this, in combination with a careful sampling scheme and network architecture Ball et al. (2023) or policy regularizations Nair et al. (2018), can turn offline data into a strong initializer for online learning. Nevertheless, these methods still require rather long online fine-tuning or high-quality offline datasets, neither of which is typically available in active positioning tasks. A more subtle scheme is to let an expert guide the data collection process, like in GuDA Corrado et al. (2024), where human-guidance is interleaved to direct trajectories toward success. Another relevant line of work is to weave online transitions into logging policies as in iterative offline RL (IORL) (Zhang et al., 2023). Here, exploratory actions are injected to discover unexplored regions in state-action space while training an offline RL agent on the generated trajectories. This approach is discussed in Section 4. Our approach is similar in spirit, but instead of exploring we want to exploit shortcuts in the trajectories to make hand-offs seamless and effective.

## 2 ACTIVE POSITIONING

In this section, we introduce the specific framework for active positioning problems building upon the framework for active alignments introduced in Burkhardt et al. (2025). There, active positioning problems are modelled as an *episodic* and *contextual* POMDP Modi et al. (2018). Specifically, the state is decomposed in the current position $s \in \mathcal{P}$ with $\mathcal{P}$ a bounded subset of $\mathbb{R}^m$ and a static context parameter $W \in \mathcal{W}$, that is $\mathcal{S} = \mathcal{P} \times \mathcal{W}$. The actions can be selected from a subset $\mathcal{A}$ of $\mathbb{R}^d$. Applying an action $a \in \mathcal{A}$ at state $(s, W)$ gives the new state $(s', W)$ with $s' = f(s, a, W)$, where $f : \mathcal{P} \times \mathcal{A} \times \mathcal{W} \to \mathbb{R}^d$ a parametrized *distortion function*. Typically, any position can be reached from any other position within one action. Note that in our scenarios, the action space is additive, meaning that the sum of two actions is itself an action if its in $\mathcal{A}$. Throughout we assume that $f(s, 0, W) = s$. Our running example is $f(s, a, W) = s + W \cdot a$ with $W \in \mathbb{R}^{d \times d}$ a distortion matrix, like a rotation matrix, but we will also consider non-linear distortions. Importantly, as $W$ stay constant throughout each episode, so is the extent of the distortion. One can think of $W$ as variances introduced by the gripping of an object, variances within an object, or conditions of the goal to be reached.

In each episode, the goal is to navigate from a random initial position $s_0$ and randomized context $W$ to a terminal state $s_W \in \mathbb{R}^d$. The reward observed at state $(s, W)$ is $R(s, a, W) = -\|f(s, a, w) - s_W\|$, i.e. the negative distance to the terminal state. An episode ends once the state is sufficiently close to $s_W$ or an upper limit of steps is reached. Formally, we terminal states are all within the set $\{(s, W) \in S : \|s - s_W\| \le \theta\}$. Typically, $W$ cannot be observed directly, often even $s$ cannot. Instead, an often high-dimensional and noised output $O_W(s) \in \mathcal{O}$ is observed, which is controlled by a conditional probability density function depending on $s$ and $W$. In total, we call the tuple $(\mathcal{P}, \mathcal{W}, \mathcal{O}, f, \gamma)$ an *active positioning problem*. This framework covers various industrial use cases, from robot arm positioning, to active alignments of optical devices (Figure 2).

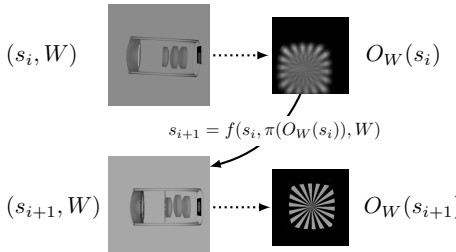

Figure 2: Active positioning of a lens system, taken from Burkhardt et al. (2025)

Although active positioning problems can also be considered as black-box optimization problems Burkhardt et al. (2025), they are inherently RL problems where symmetries and ambiguities in the need to be actively explored. For instance, the observation space is typically highly symmetric and context-dependent: states $s$ and $s'$ that are far apart can yield very similar observations $O(s, W) \approx O(s', W)$, while the same state can produce very different observations $O(s, W)$ and $O(s, W')$ under different contexts. Additionally, safety constraints and physical limitations often restrict the action space $\mathcal{A}$ so that the optimal state cannot be reached in one step and a sequence of informed actions is required. In the RL formulation, a *policy* $\pi : \mathcal{A} \times \mathcal{O} \to \mathbb{R}$ is a mapping of observations and actions to likelihood and the dynamics of the combined sys-

tem works as follows: At a given state $(s, W)$, $O(s, W)$ is observed and an action $a$ is sampled from $\pi(\cdot, O(s, W))$. The system then moves to the new state $s' = f(s, a, W)$. Note that $a$ and $s$ do not need to have same dimensionality. Starting from state $(s_0, W) \in \mathcal{S}$, the combined dynamics yields a trajectory $(s_0, W), \ldots, (s_k, W)$. The goal is to find a policy $\pi$ maximizing $J(\pi) := \mathbb{E}_{s_0, W} \left[ \sum_{i=0}^{k} -\gamma^i \|s_i - s_W\| \right]$, where $\gamma \in (0, 1)$ is a *discount factor*. Clearly, $J(\pi) = \mathbb{E}_{s_0, W}[V^\pi(s_0, W)] = \mathbb{E}_{s_0}[V^\pi(s_0)]$ with $V^\pi$ the state-value function and $V^\pi(s) := \mathbb{E}_{W \sim \mathcal{W}}[V^\pi(s, W)]$. Similarly, we define the state-action value functions $Q^\pi(s, a, W)$ and $Q^\pi(s, a)$.

## 3 THEORETICAL ANALYSIS OF SHORTCUT AUGMENTATIONS

In active positioning, good trajectories reach the optimal position in as few steps as possible. Although most logging policies used in applications visit states that are close to the optimal state, they often produce long and redundant trajectories. Our core idea is to train agents on synthetic trajectories distilled from these imperfect data, which are more direct and goal-reaching. Intuitively, we want the agent to *skip* parts of the trajectory that do not add much value — for example, going straight instead of replicating zig-zag movements or detours present in the logged data (Figure 1). However, improving logged trajectories is not straightforward. For instance, assume a collected trajectory of a logging policy contains a sub-trajectory $(s_i, W), (s_{i+1}, W), \ldots, (s_j, W)$ with actions $a_i, \ldots, a_{j-1}$, representing a long detour, like a zig-zag movement, from $s_i$ to $s_j$. Clearly, going directly from $s_i$ to $s_j$ would yield a trajectory with higher return. However, naively applying the accumulated action $a = a_i + a_{i+1} + \ldots + a_{j-1}$ at $s_i$ will not necessarily land exactly at $s_j$ due to distortions in the dynamics induced by $f$. Even small misplacements, that is ending up close to $s_j$ but not exactly at $s_j$, can cause significant value degradation if the value function is not stable in the vicinity of $s_j$. Worse, applying $'$ at $s_i$ may even move us in the opposite direction, away from $s_j$, with no guarantee that the new state has a higher value than $s_i$. Here, the length of the action $'$, the value gap between $s_i$ and $s_j$, the stability of the value function around $s_j$, and the distortion in the dynamics at $s_i$ all play a role. This section is about when the accumulated action $a$ is indeed beneficial. To start, we first formalize what it means for an action to be beneficial in our setting. We call a policy $\pi$ *distance-improving*, if for all $W \in \mathcal{W}$ we have for two subsequent states $(s_i, W)$ and $(s_j, W)$, with $i < j$ visited by the policy that $\|s_j - s_W\| < \|s_i - s_W\|$. In other words, the reward along a trajectory of $\pi$ is strictly increasing. In this section, we restrict to deterministic policies which most logging policies are. Given the deterministic transition dynamics given by $f$, the value function $V^\pi(s, W)$ is exactly the return of $\pi$ starting from $(s, W)$.

**Proposition 3.1.** *Let $\pi$ be a distance-improving policy and $(s, W), (s', W) \in \mathcal{S}$ two states on a trajectory of $\pi$ where $(s, W)$ is visited prior to $(s', W)$, then $\gamma V^\pi(s', W) - V^\pi(s, W) \geq \|s' - s_W\|$.*

All proofs are in Section A. Restricting the focus on distance-improving logging policies, beneficial actions for active positioning problems can be defined as follows:

**Definition 3.1.** *Let $\pi$ be a policy, $(s, W) \in \mathcal{S}$ a state, and $a \in \mathcal{A}$ an action with $s' = f(s, a, W)$. If $\gamma V^\pi(s', W) - V^\pi(s, W) \geq \|s' - s_W\|$, then is $a$ is a $\pi$-shortcut at $(s, W)$.*

Note that shortcuts depend on the latent information $W$, not on the position alone. The remainder of this section studies how to find shortcuts in offline trajectories. To do so, consider a short trajectory $(s_0, W), (s_1, W), (s_2, W)$ from a distance-improving policy $\pi$ with actions $a_0$ and $a_1$ (Figure 3a). In principle, an action $a$ with $s_2 = f(s_0, a, W)$ is a shortcut (Definition 3.1) and thus beneficial. However, due to distortion in $f$, we cannot assume $a = a_0 + a_1$, nor that applying $a_0 + a_1$ at $s_0$ will reach $s_2$. We must at least ensure that $a_0 + a_1$ indeed leads near $s_2$ which requires to control placement errors induced by $f$. In case $f(s, a, W) = s + W \cdot a$ with $W \in \mathbb{R}^{m \times d}$, trajectories can be augmented without placement errors:

**Proposition 3.2.** *Let $f(s, a, W) = s + W \cdot a$ and let $(s_i, W)$ and $(s_j, W)$ with $i < j$ on a trajectory of a distance improving policy $\pi$ and $a_i, \ldots, a_{j-1}$ the chain of actions $\pi$ undertook to get from $s_i$ to $s_j$. Then $a = \sum_{k=i}^{j-1} a_k$ is a shortcut for $s_i$.*

Extending Proposition 3.2 to non-linear dynamics is not trivial. Generally, we want to have that accumulating actions along a trajectory does not lead to too much placement uncertainty, which is typically the case in real-world positioning problems. We formalize this as follows:

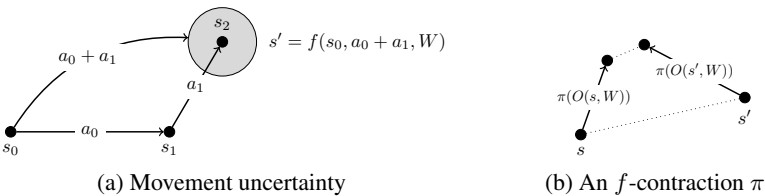

(a) Movement uncertainty        (b) An $f$-contraction $\pi$

Figure 3: Interactions of policy with movement dynamics.

**Definition 3.2** (Linear placement-errors). A distortion function $f$ has *linear placement-errors* (LPE) if there exists a constant $L_f$ such that for any chain of actions $a_0, \ldots, a_{k-1}$ with $\hat{a} := \sum_{i=0}^{k-1} a_i \in \mathcal{A}$ executed on $(s_0, W)$ with $s_i = f(s_{i-1}, a_{i-1}, W)$, we have: $\|f(s_0, \hat{a}, W) - s_k\| \leq L_f \cdot \sum_{i=0}^{k-1} \|a_i\|$.

Intuitively, the LPE property means that although a system distort movements, the mismatch introduced when regrouping actions cannot grow faster than linearly with the size of the path taken. This actually includes a wide range of functions where the distortion depends on the state only:

**Proposition 3.3.** *Let $f(s, a, W) = s + g(s, W) \cdot a$ with $g : \mathcal{S} \to \mathbb{R}^{m \times d}$ a bounded matrix-function. Then $f$ has LPE with $L_f = 2 \cdot \sup_{\mathcal{S}} \|g\|$.*

As we will see, when the distortion term also depends on the action, i.e. $g(s, a, W)$, things become more involved for small actions $a$ even if $g$ is bounded and LPE does not follow without additional assumptions (see Section 5.1.1). In Proposition B.1, we introduce an even stronger property which suffice to imply LPE for distortion functions of common active positioning problems, like linear movement dynamics. More specifically, it follows directly that a linear movement-dynamics of the form $f(s, a, W) = s + Wa$ has LPE with $L_f = 0$.

Having gathered a notion of placement errors, we now need to control the stability of the value function. Specifically, even when we can precisely reach $s_j$ from $s_i$, the value function $V^\pi$ can change drastically in the vicinity of $s_j$, making it hard to guarantee that applying the accumulated action $a$ at $s_i$ is indeed beneficial. To control this, we have to impose good prroperties on $V^\pi$. We call a value function $V : \mathcal{S} \to \mathbb{R}$ $L_V$-*Lipschitz continuous* if for all $(s, W), (s', W) \in \mathcal{S}$ we have $|V(s, W) - V(s', W)| \leq L_V \cdot \|s - s'\|$. This is the final ingredient to prove our main statement:

**Theorem 3.4.** *Let $\pi$ be distance improving and assume that $V^\pi$ is $L_V$-Lipschitz continuous and $L_f$-placement errors. Let $(s_i, W)$ and $(s_j, W)$ on a trajectory of $\pi$ and let $a = \sum_{k=i}^{j-1} a_k$ be the sum of the chain of actions $\pi$ undertook to get from $s_i$ to $s_j$. Then $a$ is a $\pi$-shortcut for $s_i$ if*

$$\gamma \cdot V^\pi(s_j, W) - V^\pi(s_i, W) - \|s_j - s_W\| \geq (\gamma \cdot L_V + 1) \cdot L_f \cdot \sum_{k=i}^{j-1} \|a_k\|.$$

In some sense, Proposition 3.2 for movement dynamics of the form $f(s, a, W) = s + W \cdot a$ arises as a special case of Theorem 3.4 because $L_f = 0$ implies that the right-hand side becomes 0 and the left-hand side is always non-negative due to Proposition 3.1. However, we note that Theorem 3.4 requires $V^\pi$ to be Lipschitz continuous, which is not needed in Proposition 3.2.

So far, we have not made any direct assumptions on policy $\pi$ beside being distance improving and $V^\pi$ being Lipschitz continuous. The next condition helps to ensure that $V^\pi$ is indeed Lipschitz continuous (see Proposition A.2 in Section A), which requires a beneficial interplay with $f$:

**Definition 3.3** ($f$-contraction). We call a policy $\pi$ an *$f$-contraction* if for all pairs $(s, W), (s', W)$ with respective observations with $o = O(s, W)$ and $o' = O(s', W)$, we have

$$\|f(s, \pi(o), W) - f(s', \pi(o'), W)\| \leq \|s - s'\|.$$

**Corollary 3.5.** *Let $\pi$ be distance improving $f$-contraction and let $f$ have LPE with constant $L_f$. Let $(s_i, W)$ and $(s_j, W)$ on a trajectory of $\pi$ and let $a = \sum_{k=i}^{j-1} a_k$ be the sum of the chain of actions $\pi$ undertook to get from $s_i$ to $s_j$. Then $a$ is a shortcut for $s_i$ if*

$$\gamma \cdot V^\pi(s_j, W) - V^\pi(s_i, W) - \|s_j - s_W\| \geq \frac{L_f}{1 - \gamma} \cdot \sum_{k=i}^{j-1} \|a_k\|$$

Being an $f$-contraction is a stronger requirement than mere distance improvement. We refer to Section B.2 for a discussion and examples of $f$-contractions and Lipschitz value functions in real-world policies. In practice, many active positioning policies do not satisfy the contraction property globally, yet this is not required for identifying useful shortcuts as shown in our experiments.

## 4 LOGGING IMPROVEMENTS VIA FINE-TUNED TRAJECTORIES

The idea of iterative reinforcement learning is to enrich logging policies with exploratory steps while collecting data (Zhang et al., 2023), mostly in order to improve coverage of the state-action space. Specifically, an *uncertainty model* $E_\theta(s, a)$ is trained with $E_\theta(s, \cdot)$ a probability distribution on $\mathcal{A}$ for each $s \in S$. Given a dataset $D$, $E_\theta$ is trained by minimizing $\mathbb{E}_{(o,a) \sim D} \big[ -\log(E_\theta(s, a)) + \mathcal{R}(\theta) \big]$ with $\mathcal{R}(\theta)$ a regularization term. Intuitively, $E_\theta(s, a)$ can be seen as the probability that action $a$ has been seen for state $s$ in $D$. Actions with small probability $E_\theta(s, a)$ at state $s$ are considered as exploratory action and should be selected according to some fixed probability $p$ enriching a given logging policy $\pi_\beta$ during rollout. These *exploratory actions* are rather rare and thus help keeping the system save and naturally close to the logging policy $\pi_\beta$ that generated the data. Although this approach seems appealing, a central part that has been underexplored in current literature, namely that static logging policies may not deal well with intermediate exploratory steps. In practise, arbitrary exploratory steps may lead to states where the logging policy cannot recover well from, leading to lower overall returns. We build upon this idea, but instead of selecting actions that have not been seen in the data, we advocate to train a $Q$-function $Q_\theta$ on some initial dataset $D$ and select actions having high $Q$-values. Formally, we set $a_\theta(s, a) = \max_{a' \in \mathcal{A}} Q_\theta(s, a')$ where $Q_\theta$ can be trained with any offline RL method, like CQL or IQL. We call $a_\theta$ an *augmentor*. By that, we aim to enrich the dataset with actions that are likely to be beneficial for $\pi_\beta$ in the sense of higher returns. While this idea is quite universal and it remains unclear how action that ease hand-overs look like in general. Moreover, in order that the augmentor provides useful steps, it has to be trained well already with limited data. The idea of LIFT to show the augmentor data of *good behavior* by applying augmentation to the logged data that emphasizes such behavior. Its easy to show that when $a_\theta$ to suggest $\pi_\beta$-shortcuts (Definition 3.1), a better logging policy can be obtained:

**Proposition 4.1.** *Let $\pi_\beta$ and $a_\theta$ be two policies, then $J(\pi_{aug}) \geq J(\pi_\beta)$ with $\pi_{aug}$ defined as follows:*

$$\pi_{\text{aug}}(O(s, W)) := \begin{cases} a_\theta(O(s, W)) & \text{if } a_\theta(O(s, W)) \text{ is a } \pi_\beta\text{-shortcut at } (s, W) \\ \pi_\beta(O(s, W)) & \text{otherwise} \end{cases} .$$

This can be seen a special case of the policy improvement theorem (Sutton & Barto, 2018, Section 4.2) to active positioning. For the remainder of this section, we discuss how to train $a_\theta$ in order that it suggests $\pi_\beta$-shortcuts for active positioning problems. However, we want to emphasize that LIFT in general is not tied to this form of backbone-augmentations.

Theorem 3.4 gives a condition when and how to augment a trajectory $(o_0, a_0, r_0), \ldots, (o_n, a_n, r_n)$ with latent states $s_i = f(s_{i-1}, a_{i-1}, W)$, observations $o_i = \mathcal{O}(s_i, W)$, rewards $r_i = -\|s_{i+1} - s_W\|$, and actions $a_i = \pi_\beta(o_i)$ from a logging policy $\pi_\beta$. To convey them into a practical algorithm, let $C \in \mathbb{R}_{\geq 0}$ be a constant and let $G_i = V^\pi(s_i, W) = \sum_{k=i}^n \gamma^{k-i} r_k$ be the returns of $\pi_\beta$. Now, take any pair $(i, j)$ with $i < j$, let $\hat{a} = \sum_{k=i}^{j-1} a_i$ be a shortcut candidate and check if $\gamma G_j - G_i + r_{j-1} \geq C \cdot \sum_{k=i}^{j-1} \|a_k\|$ with some constant $C$ holds true. Clearly, without prior information on $f$ and $\pi_\beta$, the exact value of $C$ remains unclear, and thus it has to be considered a regularization hyperparameter of our method. If $C = 0$, all pairs are considered shortcuts, if $C$ is large, only very few pairs where high reward is gained in a few short steps are considered shortcuts. If the inequality is valid for $(i, j)$, we can assume that $\hat{a}$ is a shortcut and ideally, we would add the tuple $(o_i, \hat{a}, -\|s'_j - s_W\|, o'_j)$ with $s'_j = f(s_i, \hat{a}, W)$ and $o'_j = O(s'_j, W)$ to the dataset. However, due to the movement uncertainty, there is a gap between the position $s'_j$ the shortcut leads to and the observed state $s_j$. Particularly, the image observation $O(s'_j, W)$ and the reward $-\|s'_j - s_W\|$ differ from the actually observed ones, namely $o_j$ and $r_{j-1}$. We argue, however, that in many practical applications, this gap is small, for instance if $L_f = 0$ as in linear movement dynamics $f(s, a, W) = s + W \cdot a$ (see Proposition 3.2). Thus, we add $(o_i, a, r_{j-1}, o_j)$ to the training dataset. Algorithm 1 summarizes our shortcut sampling procedure, and we want to emphasize that it can be added to any offline RL method that samples from an offline dataset, like to minimize the Bellman error or related temporal difference errors as

in CQL. Note that for a given input tuple, the runtime of Algorithm 1 is $O(n)$. Observe that the synthetic shortcuts are only used to obtain the augmentor $a_\theta$, which in turn is only used to fine-tune the logging policy, and the collected dataset consists of real data only. The precise procedure is described in Algorithm 2. For that, they must have good hand-over properties and thus we augment the dataset $D$ with shortcuts computed via Algorithm 1 when training $Q_\theta$.

---

**Algorithm 1:** Shortcut sampling

**Input**   : $C \geq 0$, $i \in [n]$, trajectory
$\{(o_0, a_0, r_0), \ldots, (o_n, a_n, r_n)\}$
**Output**  : Tuple $(o_i, \hat{a}, r_{j-1}, o_j)$
Compute returns $G_0 \ldots, G_n$ for
 trajectory
$S = \{\}$
**for** $j = i + 1 \cdots n$ **do**
  $\hat{a} \leftarrow \sum_{k=i}^{j-1} a_k$
  **if** $\gamma G_j - G_i \geq C \cdot \sum_{k=i}^{j-1} \|a_k\|$ *and*
  $\hat{a} \in \mathcal{A}$ **then**
    Add $(o_i, \hat{a}, r_{j-1}, o_j)$ to $S$
Let $m = |S|$ and denote $\hat{r} = (\hat{r}_1, \ldots, \hat{r}_m)$
 the rewards of the tuples in $S$
Let $p \sim \hat{r} - \min \hat{r}$ a mass function
Sample $(o_i, \hat{a}, r_{j-1}, o_j)$ from $S$ w.r.t. $p$
**return** $(o_i, \hat{a}, r_{j-1}, o_j)$

---

**Algorithm 2:** LIFT

**Input**   : Logging policy $\pi_\beta$, $n \in \mathbb{N}$,
augmentor $a_\theta$, $p \in [0, 1]$
**Output**  : Dataset $D$ with $n$ trajectories
**Initialize:** $D = \{\}$
**repeat**
  Sample $o_0$ from environment
  Set $d = $ false, $\tau = ()$, $i = 0$
  **while** $d$ *is false* **do**
    $a_i = \pi_\beta(o_i)$
    **if** $p' \leq p$ *with* **then**
      $a_i = a_\theta(o_i, a_i)$
      $o_{i+1}, r_i, d = $ env.step($a_i$)
      Reset $\pi_\beta$ at $o_{i+1}$ (if necessary)
    **else**
      $o_{i+1}, r_i, d = $ env.step($a_i$)
    Add $(o_i, a_i, r_i)$ to $\tau$, $i = i + 1$
  Add Trajectory $\tau$ to $D$
  **if** *train augmentor* **then**
    Train $a_\theta$ on $D$ with with Algorithm 1
**until** $|D| = n$
**return** $D$

---

## 5  EXPERIMENTS

To evaluate LIFT for active positioning problems, we address two main questions: First, can shortcut augmentations improve pure offline RL, and second, can they be leveraged during data collection by training a $Q$-based augmentor in comparison to warm-start RL? To this end, we test different distortion functions $f$, observation types $\mathcal{O}$, and levels of expertness of the logging policies.

### 5.1  ENVIRONMENTS

In order to analyze different movement distortions and observation types in isolation, we conducted our experiments in semi-realistic active positioning environments designed to keep real world characteristics and entail small sim-to-real gaps. Throughout, we use $-\|s - s_W\|$ as reward signal, which is easy to compute in simulations, as one typically has access to latent information $(s, W)$.

#### 5.1.1  MOVEMENT DISTORTIONS

We consider different movement distortions, some of them have linear forms, like $f_{\text{blend}}$ and $f_{\text{rot}}$ both with $L_f = 0$. We also use non-linear distortions, like $f_{\text{scale}}$ and $f_{\text{sin}}$ which have LPE with $L_f > 0$ and one non-continuous distortion $f_{\text{regrot}}$ also having LPE which is not contracting. We also test a movement dynamics, called $f_{\text{sqrt}}$, that does not satisfy the LPE property. We refer to Section B for their precise mathematical definitions and corresponding proofs of their properties. Figure 6 illustrates an overview of the different distortions in two dimensions.

#### 5.1.2  OBSERVATIONS

A canonical type of observation is when the position can be observed directly, i.e., $\mathcal{O}_{\text{PO}}(s, W) = s$. Here, we have to fix optimum $s_W = s^*$ throughout, as otherwise it is impossible to infer where the optimum should be without observing information about $W$. Roughly speaking, these are scenarios where it is known where the optimum is, but not how to get there. We will evaluate these scenarios in $d = 2$ and $d = 5$ dimensions. Our motivation stems from scenarios where observations are drawn

from optical sensors and hence we test our method on different image generators (Figure 4). The first comes from active alignments problems from camera assembly, were a lens objective has to be positioned relative to a sensor to obtain optimal optical performance Liu et al. (2024). Here, $s$ relates to the position of the lens objective and $W$ to variances in the lenses of the objective and distortions in the movement dynamics. At each position $s$, light is sent through the lens system creating an image $\mathcal{O}_{\mathrm{LP}}(s, W)$ on a sensor. The task is to position the objective with variances $W$ precisely to an individual optimum $s_W$ (Figure 2) As some information about $W$ is contained in the image implicitly, it is possible to design algorithms that leverage the image information to move towards $s_W$. We use the realistic generator from Burkhardt et al. (2025) where collimated light is sent in the form of a *Siemens star* producing images whose contrast and sharpness are sensitive to small misalignments, thereby providing a rich and informative signal for learning-based control.

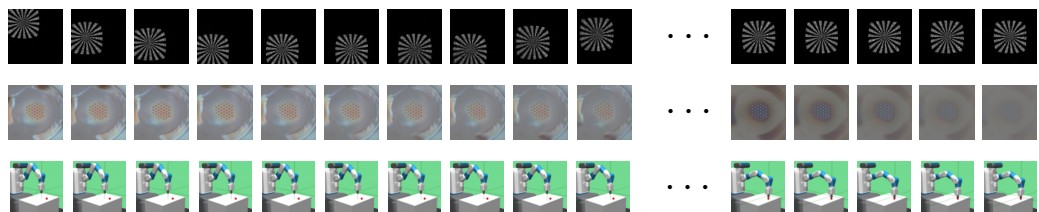

Figure 4: Exemplary trajectories of $\pi_{\mathrm{cw},l}$ executed in $\mathcal{O}_{\mathrm{LP}}$, $\mathcal{O}_{\mathrm{LT}}$, and $\mathcal{O}_{\mathrm{RI}}$ (top to bottom).

Our second image generator is the *light tunnel* from Gamella et al. (2025), where light is sent from a source through two polarizers whose angles dictate how light passes through to an optical sensor. Here, each position $s$ of the polarizers filters out certain wavelengths of the light creating a image $\mathcal{I}(s)$ at the sensor. Here, the image observation does not on depend on the context $W$ and essentially only on the relative difference of the angles of the polarizers, i.e. many states lead to the same image. To add some context, we sample in each episode $s_W$ uniformly from the box $[0, 2\pi]^2$ and set $\mathcal{O}_{\mathrm{LT}}(s, W) = \mathcal{I}(s) - \mathcal{I}(s_W)$. In our experiments, we use the decoder of the autoencoder trained on images from the real system provided in the data repository of Gamella et al. (2025).

Lastly, we run experiments in the *Fetch Reach* environment Plappert et al. (2018), where a robotic arm has to reach a position $s_W$. Here, we use the vanilla environment $\mathcal{O}_{\mathrm{RP}}(s, W) = s - s_W$ where the distance to the target is observed. In Section C we study the effect of shortcut augmentation for harder variants using image observations and reaching multiple goals subsequently.

### 5.1.3 LOGGING POLICIES

Algorithms for active positioning do not follow a general recipe, but rather depend on the specific application. Alignments of optical systems, for instance, have traditionally relied on iterative optimization of measured performance signals such as coupling efficiency or spot quality, where actuators are moved sequentially or in small patterns and the response is evaluated to guide subsequent steps, typically following coordinate-descent or heuristic search strategies that explore one or more degrees of freedom at a time (Parks, 2006; An et al., 2021; Langehanenberg et al., 2015).

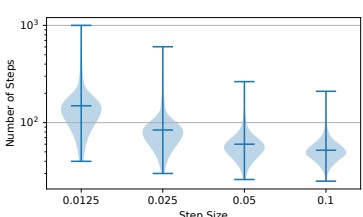

Figure 5: Expertness of $\pi_{\mathrm{cw},l}$.

Typically, the alignment starts with coarse steps and reduces the step size later, for instance (Liu et al., 2024, Section 3.1) for camera assembly. We have distilled the common principles into a synthetic logging policy called *coordinate walk*, $\pi_{\mathrm{cw},l}$ that follows a structured coordinate walk with step size $l$. This allows us to control the level of expertness of the logging policy and thus the quality of the collected data. Our synthetic position policy knows the location of $s_W$, but can only reach it via a path that is suboptimal in both, number of steps and direction. More precisely, it sequentially moves along individual coordinates of the positions $s \in \mathcal{P} \subset \mathbb{R}^m$ by choosing actions $a \in \mathcal{A}$ along unit vectors until $s_i$ matches $(s_W)_i$. Once a dimension is traversed, the policy cycles to the next coordinate and continues this procedure, thereby producing a structured, axis-aligned walk toward the optimum.

If all dimensions have been optimized, the step size $l$ is halved. By varying the initial step size, the expertness of the logging policy can be adjusted (see Figure 5). Figure 9 shows trajectories of the coordinate walk executed under different movement distortions. To model realistic hand-overs between logging policies and augmentors, we assume the internal state of the policy, i.e. the current step size $l$ and dimensions already optimized, is reset to the initial values once the policy is reset. To not make our mathematical framework introduced in Section 3 too specific for these types of resets, we assume stateless policies there. For most states, $V^{\pi_{l_2}}(s, W) \geq V^{\pi_{l_1}}(s, W)$ for two step sizes $l_1 < l_2$ holds true and thus Theorem 3.4 still hold in this specific application. In Section B.2, a detailed discussion on the contraction-property and LPE of $\pi_{\text{cw},l}$ is given.

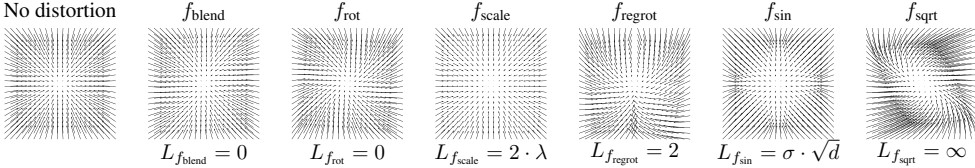

Figure 6: Movement distortions used when applying actions $\text{clip}_\lambda(s_W - s)$.

## 5.2 RESULTS

Our approach from Section 4 gives rise to essential two algorithms. First, a purely offline one that takes a static dataset collected from some logging policy and trains an offline RL algorithm with shortcut augmentations. In our experiments, we use CQL and denote this algorithm as CQL-SC. Second, an iterative offline RL algorithm that collects data with an augmented logging policy where CQL is trained on the collected data, called LIFT. If the subsequently trained CQL also uses shortcuts, we denote this algorithm as LIFT-SC. By default, we use Algorithm 2 with $p = 0.6$, limit augmentations per trajectory to 20. A detailed hyperparameter analysis is given in Section D.1.

First, we want to analyze the effect of different augmentations while collecting data and the effect of using shortcuts in the CQL training afterward. Beside naive augmentations as adding gaussian noise $\pi_\beta(o) + \epsilon$ or randomly scaling actions $\pi_\beta(o) \cdot \epsilon$ with $\epsilon = 2 \cdot \exp(\eta), \eta \sim \mathcal{N}(0, \sigma)$, we also use uniformly sampled actions from $\mathcal{A}$ and IORL-like augmentations based on an uncertainty model as in Zhang et al. (2023). We run these experiments in $(\mathcal{O}_{\text{PO}}, f_{\text{blend}})$ with step size 0.025 in $d = 5$ dimensions, collected 3 independent datasets consisting of 100 trajectories each and trained 3 independent CQL policies on each of them. The LIFT augmentor is trained once after 50 trajectories.

The averaged convergences to $s_W$ of the CQL policies, each evaluated on 20 randomly drawn contexts are shown in Figure 7a. Here, we see that, independently of shortcuts are used in the training afterward, the best CQL policies can be obtained when trained on the data collected with LIFT. Moreover, we see that when training takes place with shortcuts, every policy can be improved. This finding is underpinned when computing the dataset characteristics introduced in Schweighofer et al. (2022) shown in Figure 7b. LIFT creates trajectories having the highest average returns reproducing findings in Schweighofer et al. (2022) that this correlates with CQL performance. On the other hand, LIFT does not explore the space as good as

| $\pi_{\text{cw},l}$ | .0125 | .025 | .05 | .1 |
|---|---|---|---|---|
| $f_{\text{blend}}$ | • | • | • | • |
| $f_{\text{scale}}$ | • | • | • | • |
| $f_{\text{rot}}$ | • | • | • | • |
| $f_{\text{regrot}}$ | • | | | |
| $f_{\text{sin}}$ | • | • | • | • |
| $f_{\text{sqrt}}$ | | • | • | • |

Table 1: Cases where LIFT-SC outperforms SAC baseline in $\mathcal{O}_{\text{PO}}$, $d = 5$.

other methods, showing a clear differentiation to IORL that has been explicitly designed to explore well. However, high exploration comes at the price of an impeded hand-off back to the logging policy, leading to low trajectory qualities for IORL and random actions.

In our second type of experiments, we want to evaluate how our methods compare under different movement distortions and observation types. In $\mathcal{O}_{\text{PO}}$, algorithms collect a total of $n = 100$ and $n = 500$ trajectories for $d = 2$ and $d = 5$ respectively, where the LIFT augmentor is trained once after 50 and 100 collected trajectories respectively. In $\mathcal{O}_{\text{LP}}$, we collect 500 trajectories and LIFT is trained once after 100 episodes. In $\mathcal{O}_{\text{LT}}$, we collect only 100 trajectories and LIFT is trained once after 50 collected trajectories. Here, we additionally compare to SAC Haarnoja et al. (2018) trained

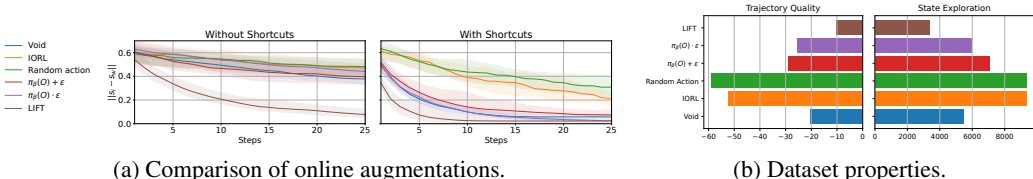

(a) Comparison of online augmentations.  (b) Dataset properties.

Figure 7: Experiments in $(\mathcal{O}_{\text{PO}}, f_{\text{blend}})$ with step size $l = 0.025$ with $d = 5$.

with a mixture of offline and online data as done in warm-start RL that is restricted to the same number of trajectories as in our offline datasets. Specifically, in a scenario with $n$ many episodes, the replay buffer of SAC is initialized with the same number of trajectories collected by the logging policy the LIFT augmentor obtains in training, e.g. $m = 50$ for $\mathcal{O}_{\text{LT}}$. Moreover, we also compare to diffusion-based techniques, like GTA Lee et al. (2024) that generate synthetic transitions and Diffusion-QL (DQL) Wang et al. (2023) that learns a diffusion-based policy. Figure 8 presents selected comparisons across the multiple scenarios and all comparisons can be found in Section D. In all tested environments, we see that CQL policies trained offline on data from LIFT have better performance than these trained on unaugmented data from the logging policy. This effect fades a bit when adding shortcuts to the subsequent offline training: In most scenarios, the performance of LIFT-SC is better or equal than CQL-SC. This is, for instance, not the case in when using image data from $\mathcal{O}_{\text{LP}}$, where CQL training on data obtained from LIFT-SC showed high variance. Studying the effect of shortcuts in isolation, CQL-SC consistently outperforms CQL and LIFT-SC consistently outperforms LIFT, making LIFT-SC the best of our methods. Comparing LIFT-SC to the SAC with offline data, we see a clear picture: SAC stays ahead in all low-dimensional cases for $\mathcal{O}_{\text{PO}}$, and LIFT-SC outperforms SAC almost consistently over all movement dynamics and expert-levels of the logging policy in $\mathcal{O}_{\text{PO}}$ for $d = 5$ (see Table 1), as well as in image-based scenarios. Interestingly, for $f_{\text{regrot}}$ where the contraction property is violated, augmentations with shortcut fail where in $f_{\text{sqrt}}$, where LPE does not hold, augmentations still help but the advantage over SAC is almost negligible.

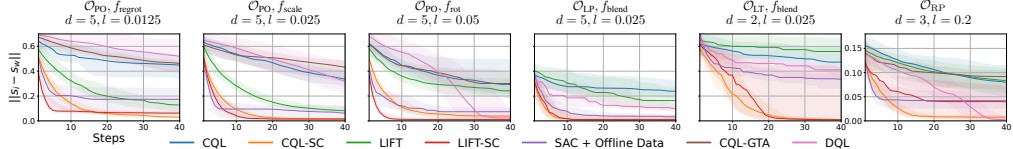

Figure 8: Comparisons of our methods under various distortions and observation types.

## 6 DISCUSSION

We demonstrate that shortcut augmentations can consistently improve the effectiveness of offline RL in active positioning problems in both, theoretical and experimental validations. In particular, we find that augmentations provide the largest gains in complex scenarios with higher action dimensionality or partial observability, where plain offline RL often fails. This suggests that exploiting task structure to expand data coverage is a promising alternative to relying solely on behavior regularization. Compared to warm-start RL, LIFT offers a more data-efficient way to leverage suboptimal expert routines: by selectively taking shortcuts suggested by an off-policy learner, we improve dataset quality without requiring extensive online fine-tuning. Nevertheless, our approach has limitations. Shortcut validity depends on assumptions about the distortion function and value function regularity, which may not hold in all real-world positioning systems. Moreover, our experiments are limited to semi-realistic simulators; future work should validate these methods on physical platforms, especially in robotic alignment tasks. Another open question is how to combine shortcut augmentation with model-based methods or world models to further improve sample efficiency. We believe that the principles underlying LIFT are broadly applicable beyond active positioning tasks where expert routines exist but are suboptimal. We hope this work encourages a more systematic treatment of data augmentation strategies for offline RL in structured industrial tasks.

ETHICS STATEMENT

This work investigates RL methods for active positioning problems, with a particular focus on data augmentation for improving offline policy learning. Our experiments are conducted exclusively in simulated environments and do not involve human subjects, personal data, or sensitive information. The proposed methods are designed for applications such as optical alignment and robotic positioning in industrial settings, where potential impacts include increased energy efficiency and reduced material waste through more accurate and reliable automation. We do not anticipate any direct negative societal consequences of this research. However, as with any advancement in machine learning for automation, care should be taken to ensure that these methods are deployed in ways that complement human expertise and respect workplace safety standards.

REPRODUCIBILITY STATEMENT

All proofs for the theoretical results in Section 3 are provided in Section A. The mathematical properties of the movement distortions used in our experiments in Section 5 are given in Section B. Further implementation details and results of all benchmarks of our experimental validation from Section 5, can be found in Section D. The implementations of our experiments are among the supplemental material of this submission and will be made available on GitHub upon acceptance.

LLM STATEMENT

Large language models (LLMs) were used to refine the manuscript's language, particularly to streamline paragraphs, improve reading flow and grammar using original drafts as input, streamlining and refining mathematical expressions, and helped to address LaTeX-specific issues. Moreover, they provided to refine the mathematical definitions of some movement distortions and polishing a lengthy proof via induction for Proposition 3.3. They also assisted in summarizing related work and provided guidance on the experimental code (e.g., refactoring and debugging hints). All outputs were reviewed and edited by the authors, who take full responsibility for the final content.

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

# A  PROOFS FOR SECTION 3

**Lemma A.1.** *Let $\pi$ be distance-improving, then $(1 - \gamma)V^\pi(s, W) \geq -\|s - s_W\|$ for all $(s, W)$.*

*Proof.* Let $(s_0, W), (s_1, W), \ldots, (s_k, W)$ be a trajectory of $\pi$ starting at $s = s_0$, then

$$
V^\pi(s, W) = -\sum_{i=1}^{k} \gamma^{i-1}\|s_i - s_W\| \geq -\|s - s_W\| \sum_{i=0}^{k-1} \gamma^i = -\|s - s_W\| \cdot \frac{1 - \gamma^k}{1 - \gamma}
$$

where we have used that $\pi$ is distance improving in every step. Finally, $(1 - \gamma)V^\pi(s, W) \geq -\|s - s_W\|(1 - \gamma^k) \geq -\|s - s_W\|$. $\qquad\square$

*Proof of Proposition 3.1.* Assume that $\tau = (s_0, \ldots, s_k)$ is the sub-trajectory of $\pi$ starting at $s = s_0$ and ending at $s' = s_k$. We prove the statement via induction on $k$. Note that since $s' \neq s$, we have $k \geq 1$. Let $k = 1$, then

$$V^\pi(s, W) = -\|s_1 - s_W\| + \gamma \cdot V^\pi(s', W)$$

and the claim holds. Now, assume the statement holds from $s_1$ to $s_k = s'$, then

$$\gamma V^\pi(s', W) - V^\pi(s_1, W) \geq \|s' - s_W\|$$

by the induction hypothesis. Furthermore, we have

$$
\begin{aligned}
\gamma V^\pi(s', W) - V^\pi(s, W) &= \gamma V^\pi(s', W) - V^\pi(s_1, W) + V^\pi(s_1, W) - V^\pi(s, W) \\
&\geq \|s' - s_W\| + V^\pi(s_1, W) - V^\pi(s, W) \\
&= \|s' - s_W\| + V^\pi(s_1, W) - (-\|s_1 - s_W\| + \gamma V^\pi(s_1, W)) \\
&= \|s' - s_W\| + (1 - \gamma)V^\pi(s_1, W) + \|s_1 - s_W\|
\end{aligned}
$$

Using Lemma A.1, we have $(1 - \gamma)V^\pi(s_1, W) + \|s_1 - s_W\| \geq 0$ and the claim follows. $\qquad\square$

*Proof of Proposition 4.1.* We denote $\pi_\beta$ simply by $\pi$ in the following. It suffices to show that the statement holds if augmentation only is applied at one single state $(\tilde{s}, W)$ as we than can apply the statement repeatedly. That is, there exists an action $a$ that satisfies:

$$\gamma \cdot V^\pi(f(\tilde{s}, a, W), W) - \|f(\tilde{s}, a, W) - s_W\| \geq V^\pi(\tilde{s}, W)$$

Let $\pi_a$ be the policy that uses $a$ at $\tilde{s}$ and on all other states coincides with $\pi$. First, we show that $J(\pi_a) \geq J(\pi)$. It suffices to show that $V^{\pi_a}(s) \geq V^\pi(s)$ for all $s \in S$. Let $(s, W)$ be an initial state. If the trajectory of $\pi$ does not traverse $\tilde{s}$, then $V^{\pi_a}(s) = V^\pi(s)$. Assume differently that the trajectory visits $\tilde{s}$ at the $t$-th step. Then, the trajectory starting at $s$ follows $\pi$ till $\tilde{s}$, then chooses the shortcut $a$, and then follows $\pi$ from $s' = f(\tilde{s}, a, W)$. The value for this trajectory is:

$$V^{\pi_a}(s, W) = V^\pi(s, W) - \gamma^t \cdot V^\pi(\tilde{s}, W) - \gamma^t \|s' - s_W\| + \gamma^{t+1} V^\pi(s', W).$$

From the assumption of $(\tilde{s}, a)$, we have

$$\gamma^t \cdot (-V^\pi(\tilde{s}, W) - \|s' - s_W\| + \gamma \cdot V^\pi(s', W)) \geq 0$$

and hence $V^{\pi_a}(s, W) \geq V^\pi(s, W)$. $\qquad\square$

*Proof of Proposition 3.2.* Since Proposition 3.1 gives that $\gamma V^\pi(s_j, W) - V^\pi(s_i, W) \geq \|s_j - s_W\|$, it is left to prove that $f(s_i, a, W) = s_j$. We have

$$f(s_i, a, W) = s_i + W \cdot \sum_{k=i}^{j-1} a_i = s_i + W \cdot a_i + W \cdot a_{i+1} + \ldots + W \cdot a_{j-1}.$$

Let $s_{i+1}, \ldots, s_{j-2}$ be the intermediate states, i.e. $s_k = f(s_{k-1}, a_{k-1}, W)$, then replacing $s_k = s_k - 1 + W \cdot a_{k-1}$ in the equation above from $k = i$ to $k = j - 1$ gives the claim. $\qquad\square$

*Proof of Proposition 3.3.* Let $a_0, \ldots, a_{k-1}$ a chain of actions and set $A = \sum_{i=0}^{k-1} = a_i$, $(s_0, W)$ an initial state and set $s_i = f(s_{i-1}, a_{i-1}, W)$. Recursively unraveling the definition of $f$ yields

$$s_k = s_0 + \sum_{i=0} g(s_i, W) \cdot a_i$$

and consequently

$$
\begin{aligned}
f(s_0, A, W) - s_k &= g(s_0, W) \sum_{i=0}^{k-1} a_i - \sum_{i=0}^{k-1} g(s_i, W))a_i \\
&= \sum_{i=0}^{k-1} (g(s_0, W) - g(s_i, W))a_i.
\end{aligned}
$$

Taking norms and using the induced matrix norm on $\mathbb{R}^{m \times d}$ gives

$$\|f(s_0, A, W) - s_k\| \leq \sum_{i=0}^{k-1} \|g(s_0, W) - g(s_i, W)\| \cdot \|a_i\|.$$

By the assumption on $g$, we have

$$\|g(s_0, W) - g(s_i, W)\| \leq \|g(s_0, W)\| + \|g(s_i, W)\| \leq 2 \cdot \sup_{S \times W} \|g\|$$

independently of the actions for all $i$ and the claim follows. $\square$

*Proof of Theorem 3.4.* For brevity, we omit $W$ in the notation of the value function. We have to show that $\gamma V^\pi(f(s_i, a, W)) - V^\pi(s_i) \geq \|f(s_i, a, W) - s_W\|$. Because $f$ has linear-placement errors, it follows directly from Definition 3.2 that $\|f(s_i, a, W) - s_j\| \leq L_f \cdot \sum_{k=i}^{j-1} \|a_k\|$ and thus

$$\|f(s_i, a, W) - s_W\| = \|f(s_i, a, W) - s_j + s_j - s_W\| \leq L_f \cdot \sum_{k=i}^{j-1} \|a_k\| + \|s_j - s_W\|.$$

On the other hand, using the Lipschitz continuity of $V^\pi$, we get

$$\gamma V^\pi(f(s_i, a, W)) - V^\pi(s_i) \geq \gamma \cdot (V^\pi(s_j) - L_V \cdot \|f(s_i, a, W) - s_j\|) - V^\pi(s_i)$$

$$\geq \gamma \cdot V^\pi(s_j) - V^\pi(s_i) - \gamma \cdot L_V \cdot L_f \cdot \sum_{k=i}^{j-1} \|a_k\|$$

Now, as the inequality from the theorem statement holds, we have

$$\gamma \cdot V^\pi(s_j) - V^\pi(s_i) \geq (\gamma \cdot L_V + 1) \cdot L_f \cdot \sum_{k=i}^{j-1} \|a_k\| + \|s_j - s_W\|$$

and plugging this into the upper equation gives the claim. $\square$

**Proposition A.2.** *Let $\pi$ be an $f$-contraction. Then $V^\pi$ is $\frac{1}{1-\gamma}$-Lipschitz continuous in the states.*

*Proof.* Define $L = \frac{1}{1-\gamma}$ and let $(s, W)$ and $(s', W)$ be two states. We prove via induction over the combined number of steps $k$ needed to reach the optimality region around $s_W$ starting at $s$ and $s'$ that

$$|V^\pi(s, W) - V^\pi(s', W)| \leq L \cdot \|s - s'\|.$$

If $k = 0$, then $s$ and $s'$ are both within the optimality region, i.e. $\|s - s_W\| \leq \theta$ and $\|s' - s_W\| \leq \theta$, then $V^\pi(s, W) = V^\pi(s', W) = 0$ and the claim holds. Now, let $o = O(s, W)$ and $o' = O(s', W)$ be the observations at $s$ and $s'$ and $s_1 = f(s, \pi(o), W)$ and $s_1' = f(s', \pi(o'), W)$ be the next states after one step of $\pi$. Particularly, the induction hypothesis holds for $s_1$ and $s_1'$, i.e. $|V^\pi(s_1, W) - V^\pi(s_1', W)| \leq L \cdot \|s_1 - s_1'\|$. Since $V^\pi(s) = -\|s_1 - s_W\| + \gamma V^\pi(s, W)$ and $V^\pi(s') = -\|s_1' - s_W\| + \gamma V^\pi(s', W)$, we have

$$|V^\pi(s) - V^\pi(s')| = |\gamma \cdot V^\pi(s_1, W) - \gamma \cdot V^\pi(s_1', W) - \|s_1 - s_W\| + \|s_1' - s_W\||$$

$$\leq \gamma \cdot |V^\pi(s_1, W) - V^\pi(s_1', W)| + |\|s_1 - s_W\| - \|s_1' - s_W\||$$

$$\leq \gamma \cdot L \cdot \|s_1 - s_1'\| + \|s_1 - s_1'\|$$

$$\leq (\gamma \cdot L + 1) \cdot \|s_1 - s_1'\|$$

$$= L \cdot \|s_1 - s_1'\|$$

where the last equation is due to $L = \frac{1}{1-\gamma}$. Finally, because $\pi$ is an $f$-contraction, we have $\|s_1 - s_1'\| = \|f(s, \pi(o), W) - f(s', \pi(o'), W)\| \leq \|s - s'\|$ and the claim follows. $\square$

*Proof of Corollary 3.5.* Because $\pi$ is an $f$-contraction, $V^\pi$ is $\frac{1}{1-\gamma}$-Lipschitz continuous by Proposition A.2. Plugging $L_V = \frac{1}{1-\gamma}$ into Theorem 3.4 gives the claim. $\square$

## B    MOVEMENT DISTORTION FUNCTIONS

In this section, we formally define the different movement distortions $f$ we consider in our experiments. The first set of distortions are linear distortions of the form $f(s, a, W) = s + W \cdot a$ with $W \in \mathbb{R}^{d \times d}$ a distortion matrix, more specific, we use

$$f_{\text{blend}}(s, a, W) = s + (I_{d \times d} + W) \cdot a, \quad W \sim \mathcal{N}_{d \times d}(0, \sigma)$$

For $W \in \mathbb{R}$ a scalar, let $R_W = \begin{pmatrix} \cos(W) & -\sin(W) \\ \sin(W) & \cos(W) \end{pmatrix}$ be a two-dimensional rotation matrix. We rise this to a high-dimensional rotation matrix where adjacent dimensions are rotated, i.e.,

$$\text{Rot}_W = \text{diag}(R_W, \ldots, R_W) \in \mathbb{R}^{d \times d}$$

where $\text{diag}(A_1, \ldots, A_k)$ is the block-diagonal matrix with blocks $A_1, \ldots, A_k$ on the diagonal.

$$f_{\text{rot}}(s, a, W) = s + \text{Rot}_W \cdot a, \quad W \sim \mathcal{N}(0, \sigma)$$

The next distortion function is a scaling-based one which does not depend on a latent context $W$:

$$f_{\text{scale}}(s, a, W) = s + \text{clip}_{C, \lambda}\left(\|s - s_W\|\right) \cdot a$$

with some constant $0 < C < \lambda$ to ensure that the steps are not to small so that the optimum can be reached in finitely many steps.

The next set of distortions is again a rotation-based one, but one where the rotation matrix depends on the region. For that, we assume the position space $\mathcal{P}$ is decomposed into $c$-many non-overlapping subsets $\mathcal{P}_1, \ldots, \mathcal{P}_c$ such that $\cup_{i=1}^{c} \mathcal{P}_i = \mathcal{P}$. Then

$$f_{\text{regrot}}(s, a, W) = s + \sum_{i=1}^{c} \mathbf{1}_{s \in \mathcal{P}_i} \cdot \text{Rot}_{W_i} \cdot a, \quad W \in \mathcal{N}_c(\mu, \sigma), \mu \in \mathbb{R}^c$$

As $\mathcal{P}_i \cap \mathcal{P}_j = \emptyset$ for $i \neq j$, only one rotation matrix is active at a time, depending on the state.

In our experiments, we used $c = 4$ and divided $\mathcal{P}$ into four sets depending on in which quadrant of $\mathbb{R}^2$ the first two dimensions reside. Moreover, we set $\mu = (-0.3, 0.6, -0.3, 0.6)$.

The next distortion is one where a non-linear offset is added which depends on both, the state and the action:

$$f_{\sin}(s, a, W) = s + a + W \cdot \sin(s) \circ \cos(s) \cdot \|a\|, \quad W \sim \mathcal{U}(0, \sigma)$$

where $\sin$ and $\cos$ are applied component-wise and $\circ$ denote the element-wise multiplication. Finally, we consider a distortion function that does not have linear placement errors:

$$f_{\text{sqrt}}(s, a, W) = s + (I_{d \times d} + W) \cdot \sqrt{\|a\|} \cdot a, \quad W \sim \mathcal{N}_{d \times d}(0, \sigma).$$

### B.1    LINEAR PLACEMENT-ERRORS

We begin by proving a stronger conditions, which is easier to check and implies LPE:

**Proposition B.1.** *Let $f$ be a distortion function and assume there exists a constant $L_f$ such that for all states $(s, W)$ and actions $a, a' \in \mathcal{A}$*

$$\|f(s, a + a', W) - f(f(s, a, W), a', W)\| \leq L_f \cdot \|a\|$$

*Then $f$ has LPE with constant $L_f$.*

*Proof.* For $i \in \{0, \ldots, k\}$, define the tail sums $\tilde{a}_i := \sum_{j=i}^{k-1} a_j$ and the states $\tilde{s}_i := f(s_i, \tilde{a}_i, W)$. By definition $\tilde{s}_0 = f(s_0, a_0 + \ldots + a_{k-1}, W)$ and, since $\tilde{a}_k = 0$ and $f(s, 0, W) = s$, we also have $\tilde{s}_k = s_k$. Thus, we have to prove that $\|\tilde{s}_0 - \tilde{s}_k\| \leq L_f \sum_{i=0}^{k-1} \|a_i\|$. Now, for any $i \in \{0, \ldots, k-1\}$ we have

$$\|\tilde{s}_i - \tilde{s}_{i+1}\| = \|f(s_i, a_i + \tilde{a}_{i+1}, W) - f(s_{i+1}, \tilde{a}_{i+1}, W)\| \leq L_f \|a_i\|.$$

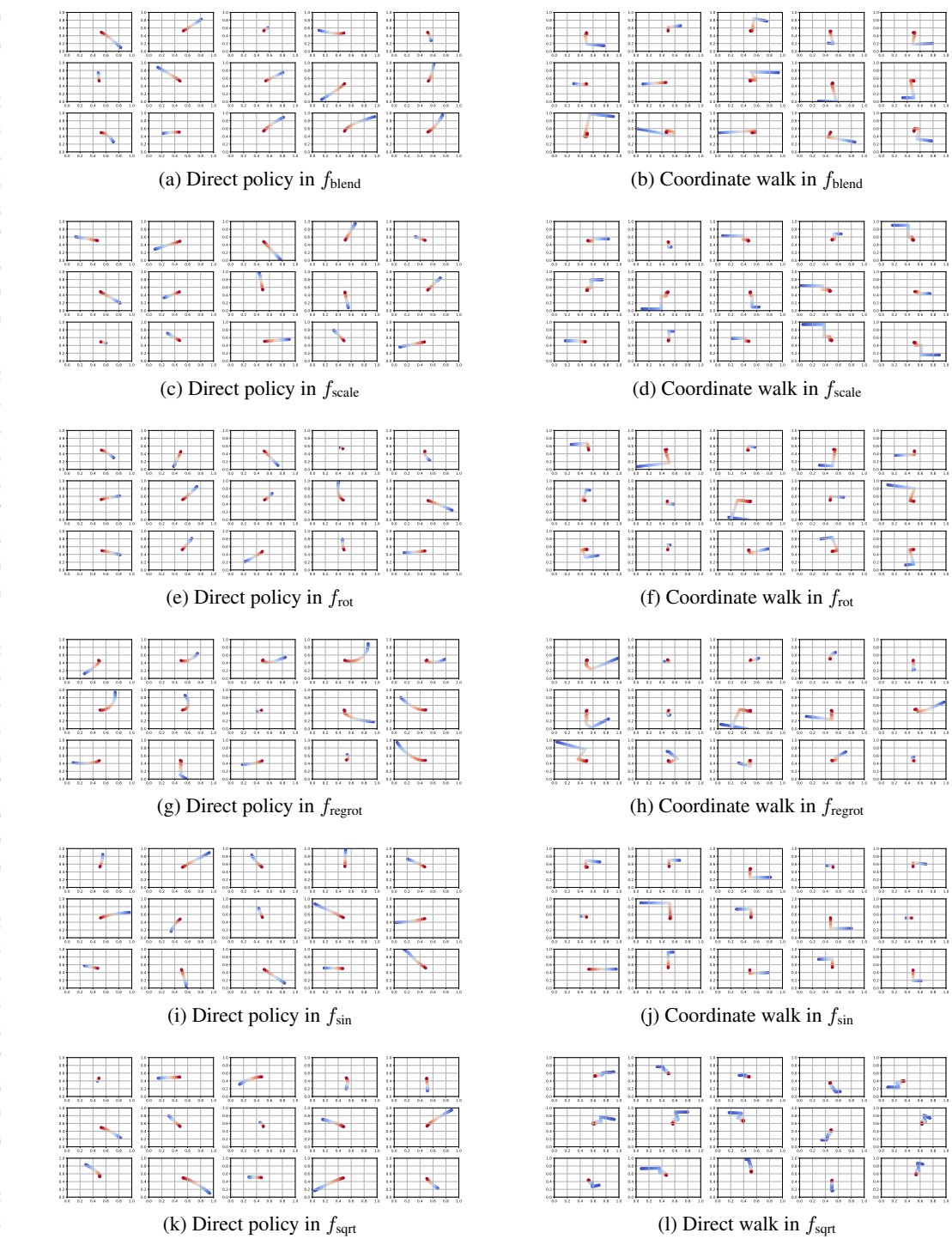

Figure 9: Trajectories of direct policy and coordinate walk in different movement dynamics.

because of the assumptions on $f$ from the statement of the proposition. Summing these inequalities and applying the triangle inequality yields

$$\|\tilde{s}_0 - \tilde{s}_k\| \leq \sum_{i=0}^{k-1} \|\tilde{s}_i - \tilde{s}_{i+1}\| \leq L_f \sum_{i=0}^{k-1} \|a_i\|.$$

$\square$

LPE and the proposition of Proposition B.1 are not equivalent: Consider $f(s,a) = s + \text{sign}(s) \cdot a$. Then its easy to show that $f$ has linear-placement errors with $L_f = 2$, but it does not have the property from Proposition B.1.

**Proposition B.2.** *The distortion $f_{\text{blend}}$ has LPE with $L_{f_{blend}} = 0$.*

*Proof.* Straight-forward application of Proposition B.1. $\square$

**Proposition B.3.** *The distortion $f_{\text{rot}}$ has LPE with $L_{f_{rot}} = 0$.*

*Proof.* Straight-forward application of Proposition B.1. $\square$

**Proposition B.4.** *The distortion $f_{\text{scale}}$ has LPE with $L_{f_{scale}} = 2 \cdot \lambda$.*

*Proof.* We write $f_{\text{scale}}(s, a, W) = s + g(s, W) \cdot a$ with $g(s, W) = \text{clip}_{C,\lambda}(\|s - s_W\|) \cdot I_d$ with $I_d$ the identity function of $\mathbb{R}^{d \times d}$. Clearly $g$ is bounded and we have $\sup_{\mathcal{S} \times \mathcal{W}} \|g\| = \lambda$ and the claim follows by an application of Proposition 3.3. $\square$

**Proposition B.5.** *The distortion $f_{\text{regrot}}$ has LPE with $L_{f_{regrot}} = 2$.*

*Proof.* We write $f_{\text{regrot}}(s, a, W) = s + g(s, W) \cdot a$ with $g(s, W) = \text{Rot}_{W_i}$ whenever $s \in \mathcal{P}_i$, where $\mathcal{P}_1, \ldots, \mathcal{P}_c$ are the partitions of $\mathcal{S}$ from Section 5.1.1. For every state $(s, W)$, $g(s, W)$ is a rotation matrix and thus $\|g(s, W)\| = 1$ and $g$ statisfies the the claim follows from Proposition 3.3. $\square$

**Proposition B.6.** *The distortion $f_{\sin}$ has LPE with $L_{f_{sin}} = \sqrt{d}\sigma$.*

*Proof.* Let $f_{\sin}(s, a, W) = s + a + g(s) \cdot \|a\|$ with $g(s, W) := W \cdot \sin(s) \odot \cos(s)$. Although we cannot apply Proposition 3.3 as $f_{\sin}$ has not the desired form, we can follow a similar strategy. First, we observe that $g$ is bounded:

$$\|g(s, W)\| = |W| \cdot \sqrt{\sum_{i=1}^{d} \sin(s_i)^2 \cdot \cos(s_i)^2} \leq \sigma\sqrt{d}$$

because $W \sim \mathcal{U}(0, \sigma)$. Let $a_0, \ldots, a_{k-1}$ be a chain of actions and set $A = \sum_{i=1}^{k-1} a_i$ and $s_i = f(s_{i-1}, a_{i-1}, W)$, then

$$f_{\sin}(s_0, A, W) - s_k = A + g(s_0, W)\|A\| - \sum_{i=0}^{k-1} \Big(a_i + g(s_i, W)\|a_i\|\Big) = g(s_0, W)\|A\| - \sum_{i=0}^{k-1} g(s_i, W)\|a_i\|$$

and thus:

$$\|f_{\sin}(s_0, A, W) - s_k\| \leq \|g(s_0, W)\|A\| + \sum_{i=0}^{k-1} \|g(s_i, W)\|a_i\| \leq \sigma\sqrt{d} \sum_{i=0}^{k-1} \|a_i\|$$

because $\|A\| \leq \sum_{i=0}^{k-1} \|a_i\|$ by the triangle inequality. $\square$

Next, we show that $f_{\text{sqrt}}$ is not LPE:

**Proposition B.7.** *The distortion $f_{\text{sqrt}}$ does not have LPE.*

*Proof.* Let $v \in \mathbb{R}^d$ be a unit vector and let $a_0 = a_1 = c \cdot v$ with $c \leq \lambda$. Let $(0, 0) \in \mathbb{R}^d \times \mathbb{R}^{d \times d}$ be an initial state, then $s_1 = f_{\text{sqrt}}(0, a_0, 0) = \sqrt{c} \cdot c \cdot v$ and $s_2 = f_{\text{sqrt}}(s_1, a_1, 0) = 2\sqrt{c} \cdot c \cdot v$. Moreover, we have $f(s_0, a_0 + a_1, 0) = f(0, 2 \cdot c \cdot v, 0) = 2\sqrt{2c} \cdot c \cdot v$ and hence

$$\|f(s_0, a_0 + a_1, 0) - s_2\| = (2\sqrt{2} - 2) \cdot \sqrt{c} \cdot c.$$

which cannot be bounded by $L_f \cdot (\|a_0\| + \|a_1\|) = 2 \cdot L_f \cdot c$ for any constant $L_f$. $\square$

## B.2 CONTRACTIONS AND LIPSCHITZ-CONTINUITY IN REAL-WORLD APPLICATIONS

We do not expect that policies and distortions from real-world applications satisfy the rigorous mathematical assumptions stated in Section 3. Pedantically, even simple modeling choices already break global smoothness: for instance, having $\mathcal{A} = B_\lambda(0)$ with $\mathcal{A}$ a strict subset of $S$, combined with an optimality region defined by a threshold $\theta$, induces discontinuities in the value function. The same holds for the coordinate walk policy in Section 5.1.3, where a fixed step length produces value functions with sharp discontinuities, as shown in Figure 11.

Nevertheless, global mathematical rigor is not required to detect local shortcuts in real trajectories. A striking example is the coordinate walk under $f_{\text{regrot}}$: since different rotations apply in different regions, the policy is not an $f$-contraction globally, because nearby states $s$ and $s'$ lying in different regions $\mathcal{P}_i$ and $\mathcal{P}_j$ may be rotated in different directions (Figure 10a). Yet, for states within same region where the coordinate walk applies same actions, the contraction property is preserved (Figure 10b). This illustrates that shortcut identification relies less on global guarantees and more on local structure along trajectory segments.

Informally speaking, it suffices that the value function does not change too abruptly for small misplacements, so that local improvements can be exploited as shortcuts. In practice, this condition is often met: physical systems typically exhibit continuity over small ranges of motion, even if discontinuities or non-contractive behavior emerge globally. Hence, while our theoretical assumptions provide clean guarantees, the underlying ideas remain applicable well beyond the idealized setting as demonstrated by our experiments in Section 5.

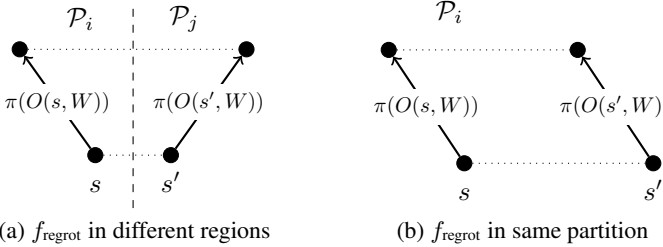

(a) $f_{\text{regrot}}$ in different regions      (b) $f_{\text{regrot}}$ in same partition

Figure 10: In $f_{\text{regrot}}$, starting at two close-by states $s$ and $s'$ in different regions $\mathcal{P}_1$ and $\mathcal{P}_2$ can increase the distance between subsequent states as opposed rotation matrices apply.

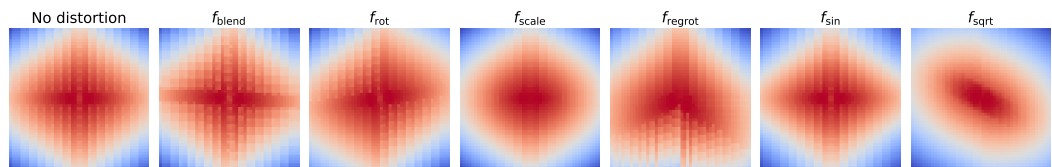

Figure 11: Value functions $V^\pi(\cdot, W)$ of coordinate walk for a random but fixed context $W$ each.

## C ADDITIONAL EXPERIMENTS IN FETCH-ENVIRONMENT

In extension to the reach experiments in Section 5 where the positional differences are directly observed, we provide in this section a proof of principle that shortcut augmentations can also benefit offline RL methods in more involved robotic environments. To this end, we consider two scenarios based on the Fetch environment Plappert et al. (2018). In the first scenario, we study a reaching task in which the robotic arm must reach a target position in 3D space. The observation is an image of the scene. We collect 100 trajectories using the coordinate walk policy described in Section 5.1.3.

In the second scenario, we consider a variant of the pick-and-place task where the robotic arm must move an object from a random initial position to a random target position. We focus solely on the positioning, i.e., the object does not need to be grasped, only touched, assuming perfect gripper

control. The policy used here performs two consecutive coordinate walks: one to reach the object and one to reach the target position. The observations are given by the distances from the gripper to the object and from the gripper to the target where the first distance is zeroed once solves the touching task. In this setting, we collect 1000 trajectories. On the collected datasets, we train CQL both with and without shortcuts, and the results are reported in Figure 12.

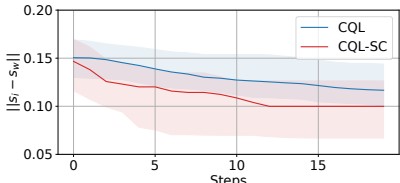 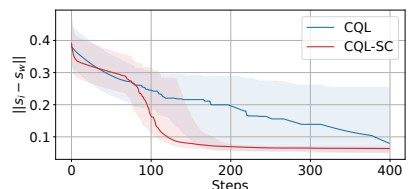

(a) Image-based reaching in $d = 3$ with 100 trajectories

(b) Position based pick-and-place in $d = 3$ with 1000 trajectories

Figure 12: Experiments in the Fetch environment.

.

# D    DETAILS FOR EXPERIMENTAL RESULTS

## D.1    HYPERPARAMETERS OF LEARNING ALGORITHMS

| Parameter | Value | Parameter | Value | Parameter | Value |
|---|---|---|---|---|---|
| actor learning rate | $10^{-3}$ | actor learning rate | $10^{-3}$ | actor learning rate | $10^{-3}$ |
| critic learning rate | $10^{-3}$ | critic learning rate | $10^{-3}$ | critic learning rate | $10^{-3}$ |
| conservative weight | 5.0 | conservative weight | 5.0 | batch size | 256 |
| $\alpha$-threshold | 10.0 | $\alpha$-threshold | 10.0 | n updates per step | 5 |
| batch size | 500 | batch size | 500 | n critics | 2 |
| $\gamma$ | 0.99 | $\gamma$ | 0.99 | $\gamma$ | 0.99 |
| $\tau$ | 0.005 | $\tau$ | 0.005 | $\tau$ | 0.005 |

Table 2: Parameter for CQL trained on collected datasets.

Table 3: Parameter for CQL trained as LIFT augmentor.

Table 4: Parameter for SAC.

## D.2    HYPERPARAMETER STUDY OF LIFT

In this section, we study effects of the different hyperparameters of the shortcut computation (Algorithm 1) and LIFT (Algorithm 2). First, we study the effect of the number of augmentations per trajectory $n$ and the probability of applying an augmentation $p$. The results are shown in Figure 13. One can see that as few as 20 augmentations per trajectory are sufficient to achieve a substantial improvement in performance, provided that the augmentation probability is not too low. Notably, higher probabilities correspond to augmentations being applied earlier in the trajectory. This suggests that augmentations at the beginning of a trajectory are more beneficial than those applied later.

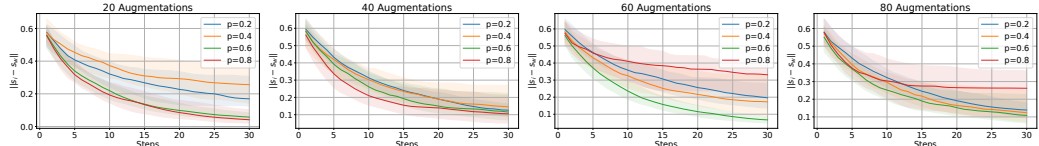

Figure 13: Experiments in $f_{\text{blend}}$ with step size 0.025 and different probabilities $p$ of applying augmentations and different maximal number of augmentations per trajectory

Next, we analyse the effect of the sampling scheme of shortcuts along a trajectory. Here, we denote the sampling mechanism described in Algorithm 1 as *weighted*. Another way to sample shortcuts

from the set $S$ computed in Algorithm 1 is to use a distribution that is proportional to the inverse distance to the optimum, i.e. $p(i) \sim \frac{1}{\|s_i - s_W\|}$ or to sample uniformly from $S$. Instead of sampling, one can also just use the shortcut residing within the action space that leads to the point of highest reward within the trajectory called *best*. The results are shown in Figure 14 for $n = 20$ augmentations per trajectory and $p = 0.4$ showing that in the environments we consider, the sampling strategy does not have a significant effect on the performance.

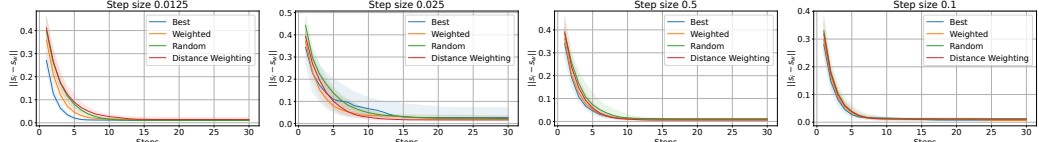

Figure 14: Experiments in $f_{\text{blend}}$ with different step size and different sampling strategies.

## D.3 ADDITIONAL VISUALIZATION

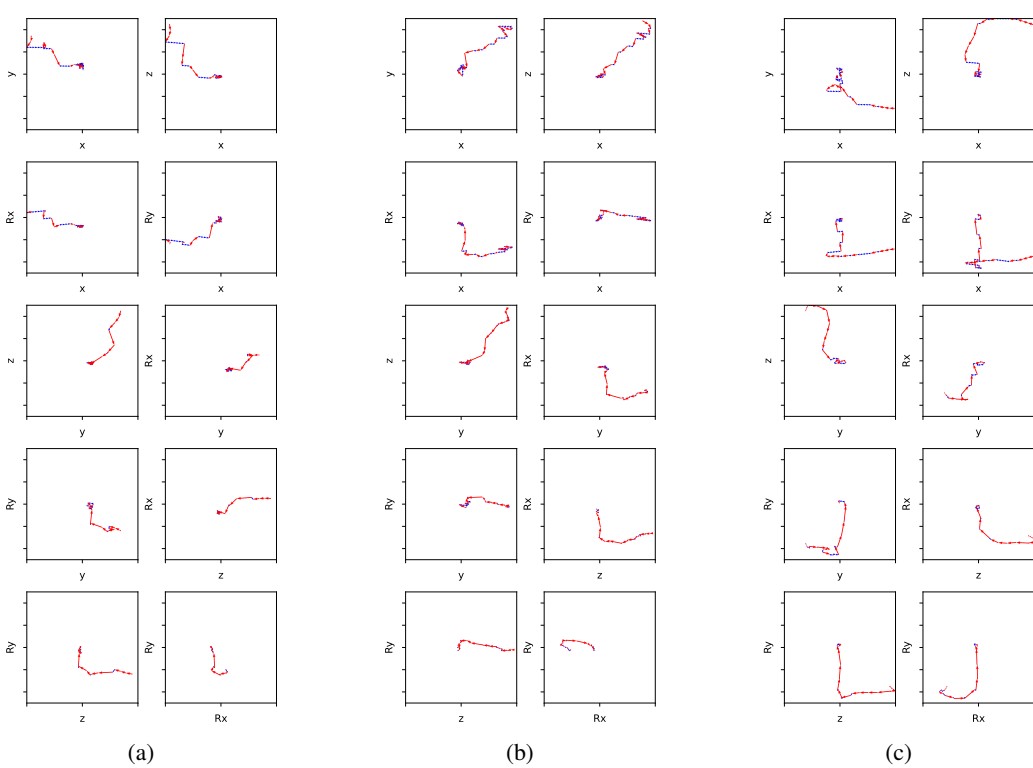

(a)          (b)          (c)

Figure 15: Augmented trajectories generated by LIFT for $\mathcal{O}_{\text{LP}}$ in 5 dimensional hidden position space: Actions coming from the augmentor in red and actions from the logging policy in blue.

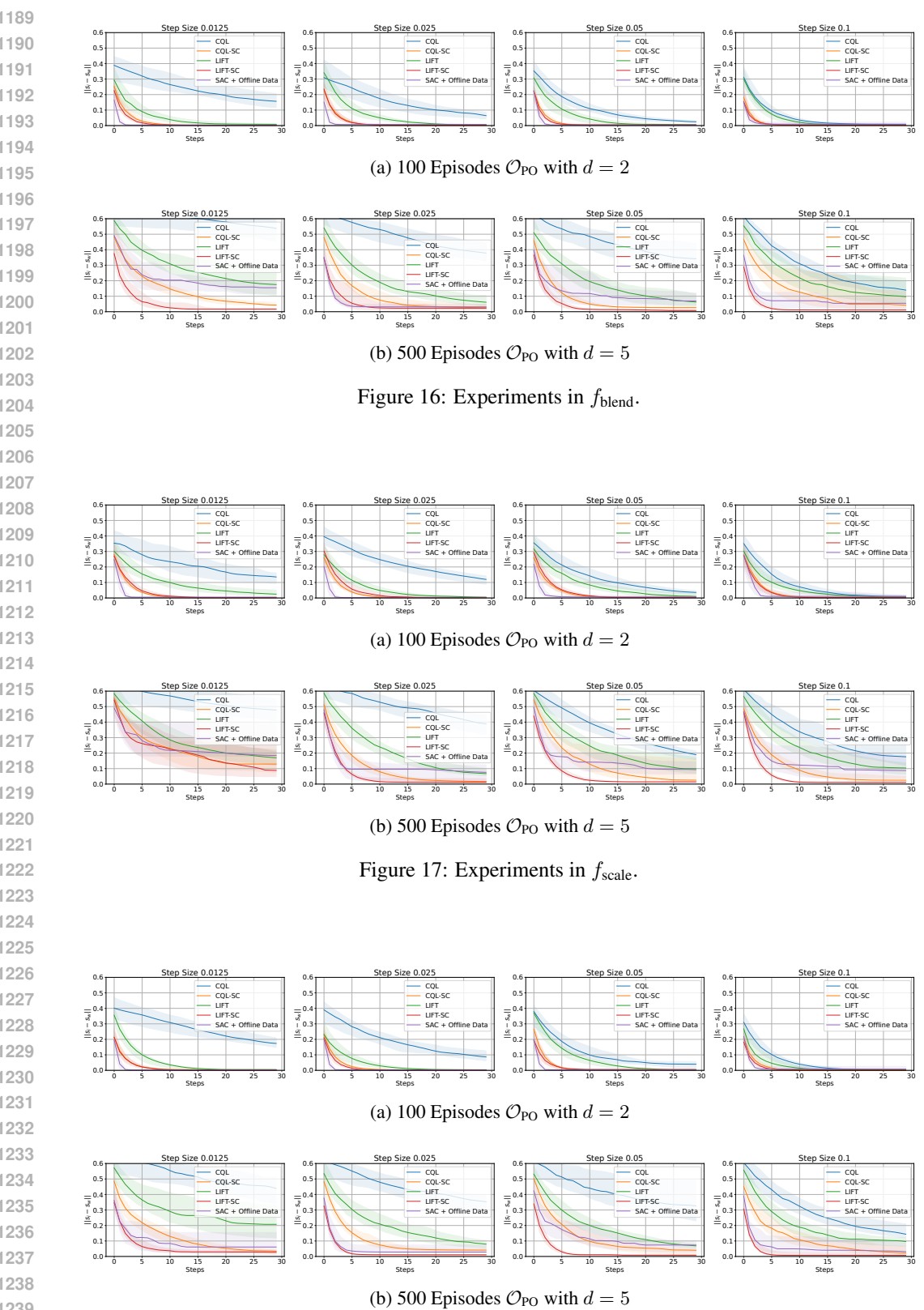

(a) 100 Episodes $\mathcal{O}_{\text{PO}}$ with $d = 2$

(b) 500 Episodes $\mathcal{O}_{\text{PO}}$ with $d = 5$

Figure 16: Experiments in $f_{\text{blend}}$.

(a) 100 Episodes $\mathcal{O}_{\text{PO}}$ with $d = 2$

(b) 500 Episodes $\mathcal{O}_{\text{PO}}$ with $d = 5$

Figure 17: Experiments in $f_{\text{scale}}$.

(a) 100 Episodes $\mathcal{O}_{\text{PO}}$ with $d = 2$

(b) 500 Episodes $\mathcal{O}_{\text{PO}}$ with $d = 5$

Figure 18: Experiments in $f_{\text{rot}}$.

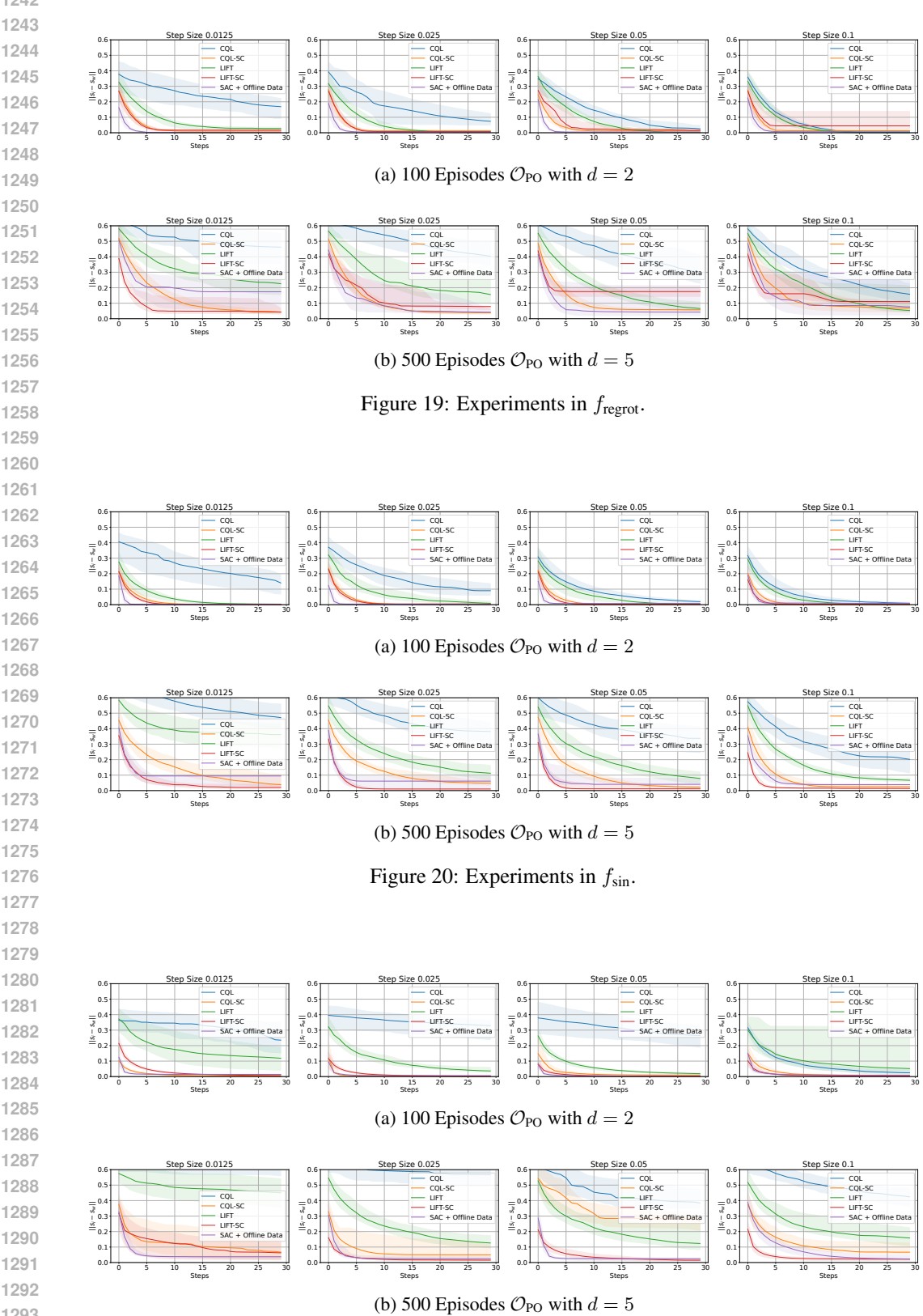

(a) 100 Episodes $\mathcal{O}_{\text{PO}}$ with $d = 2$

(b) 500 Episodes $\mathcal{O}_{\text{PO}}$ with $d = 5$

Figure 19: Experiments in $f_{\text{regrot}}$.

(a) 100 Episodes $\mathcal{O}_{\text{PO}}$ with $d = 2$

(b) 500 Episodes $\mathcal{O}_{\text{PO}}$ with $d = 5$

Figure 20: Experiments in $f_{\text{sin}}$.

(a) 100 Episodes $\mathcal{O}_{\text{PO}}$ with $d = 2$

(b) 500 Episodes $\mathcal{O}_{\text{PO}}$ with $d = 5$

Figure 21: Experiments in $f_{\text{sqrt}}$.

