# OpenReview forum: "Augmentations in Offline Reinforcement Learning for Active Positioning"
_ICLR.cc/2026/Conference — Submitted to ICLR 2026_

### Official Review · Reviewer_9oTn · 2025-10-30

**Soundness:** 2
**Presentation:** 1
**Contribution:** 1
**Rating:** 2
**Confidence:** 3

**Summary:**

The paper proposes LIFT, a data augmentation method for offline reinforcement learning specifically designed for active positioning problems. The authors provide theoretical justification for LIFT and demonstrate empirically that they improve data quality and the performance of the final offline RL policy.

**Strengths:**

1. Practical impact on active positioning

LIFT makes offline RL practically useful for active positioning, delivering consistent improvements over baselines.


2. Theoretical Support

Rather than relying solely on empirical results, the paper mathematically establishes the effectiveness of the LIFT framework.

**Weaknesses:**

1. Missing related works of data augmentation for offline RL

A significant limitation of this paper's empirical evaluation is the omission of relevant baselines for data augmentation in offline RL. The community has extensively explored this area, with established lines of work including noise injection techniques [1, 2] and, more recently, generative models [3, 4, 5]. Without comparisons against these crucial and relevant methods, it is difficult to assess the relative effectiveness of LIFT.

2. Narrow applicability of the LIFT

While LIFT demonstrates commendable results on the active positioning problem, its contribution to the broader machine learning community appears limited. The methodology is presented as a combination of domain-specific heuristics, tightly coupled to the particularities of the active positioning task. The paper does not sufficiently demonstrate the generalizability of its approach or abstract the core techniques into a framework that would be insightful for other problems. Given ICLR's focus on foundational and broadly applicable research, the narrow focus of this work is a significant concern.

[1] Laskin, Misha, et al. "Reinforcement learning with augmented data." Advances in neural information processing systems 33 (2020): 19884-19895.

[2] Sinha, Samarth, Ajay Mandlekar, and Animesh Garg. "S4rl: Surprisingly simple self-supervision for offline reinforcement learning in robotics." Conference on Robot Learning. PMLR, 2022.

[3] Lu, Cong, et al. "Synthetic experience replay." Advances in Neural Information Processing Systems 36 (2023): 46323-46344.

[4] Lee, Jaewoo, et al. "Gta: Generative trajectory augmentation with guidance for offline reinforcement learning." Advances in Neural Information Processing Systems 37 (2024): 56766-56801.

[5] Li, Guanghe, et al. "Diffstitch: Boosting offline reinforcement learning with diffusion-based trajectory stitching." arXiv preprint arXiv:2402.02439 (2024).

**Questions:**

Offline RL methods, such as CQL and IQL, mentioned in the paper, are relatively outdated, and significant advancements have been made in the field of offline RL. In particular, policies utilizing diffusion models [6, 7] have demonstrated powerful performance on various offline RL benchmarks such as D4RL [8] and OGBench [9]. It would be interesting to investigate whether diffusion-based policies can work effectively for the active positioning problem without requiring any augmentation.

[6] Wang, Zhendong, Jonathan J. Hunt, and Mingyuan Zhou. "Diffusion policies as an expressive policy class for offline reinforcement learning." arXiv preprint arXiv:2208.06193 (2022).

[7] Dong, Zibin, et al. "Cleandiffuser: An easy-to-use modularized library for diffusion models in decision making." Advances in Neural Information Processing Systems 37 (2024): 86899-86926.

[8] Fu, Justin, et al. "D4rl: Datasets for deep data-driven reinforcement learning." arXiv preprint arXiv:2004.07219 (2020).

[9] Park, Seohong, et al. "Ogbench: Benchmarking offline goal-conditioned rl." arXiv preprint arXiv:2410.20092 (2024).

---

> ### Author Response · Authors · 2025-11-21
>
> We thank you for your valuable feedback and for your pointers to diffusion-based methods that help
> to put our work into context.
>
> We discussed the applicability of our methods in [our general response](https://openreview.net/forum?id=AsVH1FQGuR&noteId=APbALO6G1M). While the scope of our
> empirical study is focused, it also comes with theoretic results underpinning our evaluation, and we
> do not see a conflict with ICLR’s scope. Nevertheless, we currently work on extending our methods to
> concrete robotic tasks as mentioned in our general response and we appreciate your suggestions on
> that matter.
>
> Regarding the extension of our comparisons: We finished a first round of experiments with
> diffusion-based policies and with diffusion-based augmentations as you suggested. You can find the
> extended comparisons with the existing environments in Figure 7. We appreciate feedback from you on
> these new results and we are happy to follow up with more experiments if needed.

---

### Official Review · Reviewer_Paah · 2025-10-31

**Soundness:** 2
**Presentation:** 2
**Contribution:** 3
**Rating:** 4
**Confidence:** 2

**Summary:**

This work describes two methods for improving offline RL learning. First, is a strategy for data augmentation that uses geometric properties of the world to improve trajectories. Second, is an injection of new actions when collecting data with the expert trajectory generator. They provide a a theoretical definition of their first method which they call shortcuts. At a high level it uses relative actions on a position (like an end effector of a robot) to augment trajectories. They run experiments on both of their methods.

Some typos:
014 - "and quantify the effects..." Should be quantifies probably
036 - promise is -> promises to

**Strengths:**

The idea is interesting here and improving sample efficiency with data augmentation is promising.

The general principle here and the specific ideas discussed in the paper seem reasonable and interesting.

**Weaknesses:**

The writing is hard to follow and unclear at times. For example the intro is sort of confusing. I think this is the main issue with the paper. It seems like a good paper but it is hard to follow and understand the setup, experiments and motivation.

Should probably cite works like this: https://arxiv.org/pdf/2310.18247

Theoretical analysis is important and seems correct but I don't understand the purpose of why we need it in this case. The general ideas presented seem to be straightforward and I don't think the theoretical section is well motivated. This may just be a writing problem but it feels out of place and too lengthy, some more intuition and motivation would go a long way.

The empirical section isn't very convincing. The environments section isn't descriptive enough in my opinion. The results themselves seem decent but it is just difficult to understand the setup.

All of the figures in the paper are small and hard to read.

**Questions:**

062 - what is a policy-time augmentation?

---

> ### Author Response · Authors · 2025-11-18
>
> ## Presentation
>
> We thank you for the feedback on the clarity of the writing. We are sorry that the current version
> made it difficult for you to follow the problem setup, motivation, and experiments. As you are the
> only reviewer wo raised concerns about the clarity of the introduction, we appreciate specific
> suggestions on which parts were particularly confusing to you. We will then revise the manuscript
> based on your feedback.
>
> Regarding the setup in the experiment section: Due to space constraints, we could only briefly
> introduce the experimental setup and we refered to the prior works for more details. For instance,
> the scenario LP (Lens Positioning) is described in reference Burkhardt et al. (2025). However, we
> agree with the reviewer that more details here would be helpful and make the paper more
> self-contained, thus we will add more details about our setups. For instance, we decided to add a
> visualization of positionings for lens systems prominently in Section 2. Also here, we would highly
> appreciate specific suggestions which parts of our setup there need more explanations.
>
> ## Theoretical analysis
>
> Reviewer `j18R` also asked for a more intuitive explanation of the theoretical arguments and we hope
> our walkthrough in the response given in the response there helps understanding why our theoretical
> investigations are required. We are happy for feedback if this helps understanding the motivation
> better.
>
> ## Related work
>
> We appreciate the pointer to GuDA and we will discuss similarities and differences in the related
> work section.

---

### Official Review · Reviewer_edFm · 2025-10-31

**Soundness:** 3
**Presentation:** 3
**Contribution:** 3
**Rating:** 6
**Confidence:** 3

**Summary:**

This paper presents LIFT (Logging Improvement via Fine-Tuned Trajectories), a framework for data augmentation in offline RL applied to active positioning tasks, such as optical or robotic alignment. The key idea is to exploit the geometric structure of positioning tasks to derive “shortcut augmentations”, i.e., trajectory-level perturbations that preserve the underlying task geometry while improving sample diversity and policy support. The authors propose two complementary modes of augmentation: 1) Static trajectory augmentation, which uses structure-aware perturbations (shortcuts) derived from the geometry of the transition dynamics and value functions. 2) Policy-time augmentation, which injects optimistic off-policy actions into the logging process, guided by a Q-function trained on augmented data. They derive theoretical guarantees for when these shortcuts improve expected return, under assumptions of Lipschitz continuity of the value function, linear placement error (LPE) in the transition function, and f-contraction of the policy. Empirically, the method is validated on synthetic and semi-realistic active positioning environments with various movement distortions and observation models (e.g., 2D/5D positional data, optical image generators). Results show that LIFT and its variant LIFT-SC (with shortcut-augmented CQL training) improve data quality and final policy performance over CQL, IORL, and warm-start SAC baselines.

**Strengths:**

The authors establish a connection between trajectory perturbations, geometry of value landscapes, and movement dynamics, grounding shortcut augmentations in formal RL theory. The derived theorems provide interpretable conditions under which shortcuts are guaranteed to improve performance.

**Weaknesses:**

1) The theoretical results rely on Lipschitz continuity, f-contraction, and linear placement errors (LPE), assumptions that may not hold in more complex, discontinuous real-world systems (e.g., frictional or hysteretic actuators). The paper acknowledges this but does not propose methods for verifying or relaxing these assumptions.

2) Experiments are conducted in semi-simulated optical systems. While these are realistic, validation on a physical robotic or optical alignment platform would greatly strengthen the paper’s impact and credibility.

3) Shortcut sampling (Algorithm 1) requires O($n^2$) pairwise comparisons within trajectories. Although feasible for short logs, this may become expensive in longer sequences or higher-frequency data.

**Questions:**

Q1: The theoretical results presume accurate $V_{\pi}$. In practice, when $V_{\pi}$ is estimated from noisy data or with function approximation, how robust is Algorithm 1 to error propagation in shortcut detection?

Q2: The augmentor is trained using synthetic shortcuts to guide real data collection. Given the policy–Q coupling, does this training exhibit instability akin to standard off-policy actor–critic divergence?

Q3: Many domains (e.g., autonomous driving, manipulation) exhibit structured dynamics and suboptimal experts. Could the proposed framework generalize beyond additive action spaces and static contexts?

---

> ### Author Response · Authors · 2025-11-20
>
> We thank the reviewer for the helpful comments and questions. While we do not have complete answers
> to all points, we try to provide our current understanding and insights below.
>
> ## Generalization beyond Lipschitz continuity
>
> At this stage, we do not have a clear path to formally relaxing the Lipschitz continuity assumption.
> However, we believe that requiring global Lipschitz continuity is likely stronger than necessary in
> practice. Notably, in several of our experiments we deliberately violate some of the theoretical
> assumptions (e.g., contraction/LPE conditions on f), yet the method still delivers strong empirical
> performance. This suggests that the approach may only require weaker, more local regularity
> conditions.
>
> ## Computational complexity of shortcut sampling
>
> The computational complexity of shortcut sampling is O(n) in the trajectory length n. We first
> sample a start index and then scan forward to select an endpoint, which in total requires at most 2n
> operations per trajectory. We have added a brief remark on this point in the revised version.
>
> ## Answers to specific questions
>
> ### Q1
> We do not have a precise theoretical characterization here. Algorithm 1 does not rely on a learned
> value function or Q-function; it operates solely on empirical returns computed from logged
> trajectories. For this reason, we expect shortcut detection itself to be comparatively robust to
> function-approximation error, with the main source of noise coming from Monte Carlo return
> estimation rather than from a parametric critic.
>
> ### Q2
>
> In our experiments, the augmentor is obtained by training a (pessimistic) CQL agent on data that has
> been augmented with synthetic (optimistic) shortcuts. Thus, we think the training loop is much closer to offline
> Q-learning with pessimism than to a fully coupled off-policy actor–critic scheme.
>
> ### Q3
>
> Our theoretical analysis of shortcuts currently relies on additive action spaces and, to some
> extent, on static contexts. However, we believe that the core idea behind LIFT (Algorithm 2) —
> namely, inserting optimistic actions into collected trajectories via an augmentor — can be
> generalized beyond this setting (see also our comment on the applicability in [our general
> response](https://openreview.net/forum?id=AsVH1FQGuR&noteId=APbALO6G1M)). In particular, we expect
> the framework to extend to more structured domains, such as those mentioned by the reviewer,
> provided one can define suitable "shortcut"-actions or primitives and mild local regularity
> conditions.

---

### Official Review · Reviewer_j18R · 2025-11-01

**Soundness:** 3
**Presentation:** 1
**Contribution:** 2
**Rating:** 2
**Confidence:** 2

**Summary:**

This paper proposes LIFT, a framework for enhancing offline reinforcement learning by augmenting logged trajectories through learned “shortcuts.” LIFT has two core features: (1) a trajectory-level augmentation that replaces suboptimal action subsequences with shorter, higher-value transitions, and (2) a policy-time augmentation, which intermittently injects high-value actions during data collection.
Theoretical analysis provides sufficient conditions under which these shortcut augmentations provably improve performance.
Experiments on simulated positioning tasks show that LIFT and its shortcut variant (LIFT-SC) outperform standard offline RL baselines.

**Strengths:**

1. The problem is interesting and properly motivated.
1. The paper provides useful ablation. Ablations vary distortions (linear and non-linear, with and without LPE), observation models (state vs. image), logger expertness, and dimensionality, plus analyses of augmentation frequency/probability and shortcut sampling schemes.

**Weaknesses:**

I currently vote to reject, though I hope to discuss potential misunderstandings with the authors during the rebuttal period.

# Experiments

The experiments appear narrowly focused on a highly specialized class of problems, characterized by a particular reward structure and transition dynamics. While the reported results are consistent and demonstrate that LIFT outperforms relevant baselines within this setup, the scope of applicability remains somewhat limited. It is unclear whether the observed gains would transfer to more general offline RL settings where the assumptions underlying LIFT may not hold. Expanding the evaluation to include at least one less structured domain (e.g., a standard offline RL benchmark or a stochastic environment with partial observability) would strengthen the paper’s empirical claims and help demonstrate that LIFT’s benefits are not confined to this narrow task family.

# Theory

I found the theoretical section somewhat difficult to follow, particularly because it was not clear why the theoretical machinery was introduced or how it would later support the proposed method. Providing additional context on the purpose of these definitions and propositions (e.g., whether they justify shortcut validity, guarantee improvement, or simply formalize intuitive conditions) would help orient the reader.

It would also greatly improve readability to include intuitive explanations alongside the formal statements. For example, some of the definitions appear to correspond to well-known RL concepts, but this connection is not made explicit. In particular, I was initially confused by Definition 3.1 and the following propositions, but on closer inspection, they seem to express a familiar idea: that a “$\pi$-shortcut” is an action a that yields higher return than an action sampled from the policy $\pi$---in other words, an action with positive advantage. Formally, this interpretation follows from the inequality in Definition 3.1:

$$\gamma V^{\pi}(s', W) - V^{\pi}(s, W) \ge \| s' - s_W \| \ge 0$$

We can rewrite this as

$$-\| s' - s_W \| \gamma V^{\pi}(s', W) \ge V^{\pi}(s, W)$$

and then recognizing that $-\| s' - s_W \| $ is the reward for taking action $a$ in state $s$, we can write

$$r(s,a) + \gamma V^{\pi}(s', W) \ge V^{\pi}(s, W)$$

which is just the TD-error, an unbiased estimate of the advantage under $pi$. Proposition 3.2 then immediately follows, because choosing an action with positive advantage is by definition better than sampling from $\pi$.

## Related Works

> it remains unclear how the data-generating logging policy limits what an offline learner can achieve.

Many prior works have studied the importance of high-coverage and near-expert quality data for offline RL. For instance, Corrado et al [1] and Kumar et al [2] emphasize the importance of expert data, while Yarats et. al emphasize data diversity / coverage. Works such as these are worth discussing in the related work.

There's also a rich literature in data augmentation for non-visual, state-based RL tasks that is not discussed [e.g, 1, 4-7]. For instance, Van de Pohl [5] and Corrado & Hanna [5]  define different classes of augmentations that leverage symmetries in an environment. Corrado et. al [1] referenced in the preceding paragraph also leverages symmetries to generate augmented data. Pitis et. al [6] provides a general framework that captures a lot of these augmentations.

> This illustrates that shortcut identification relies less on global guarantees and more on local structure along trajectory segments.

Pitis et. al [6,7] discuss this idea as well. They observe that causal independences in a task's features are not global but local, and use this information to generate augmented data.

1. Corrado et. al. Guided Data Augmentation for Online Reinforcement Learning and Imitation Learning. RLC 2024. https://arxiv.org/abs/2310.18247
2. Kumar et. al. When Should We Prefer Online Reinforcement Learning Over Behavioral Cloning? ICLR 2022. https://arxiv.org/abs/2204.05618
3.  Yarats et. al. Don’t Change the Algorithm, Change the Data: Exploratory Data for Online Reinforcement Learning. https://arxiv.org/abs/2201.13425
4. Van de Pol et. al. MDP Homomorphic Networks: Group Symmetries in Reinforcement Learning. https://arxiv.org/abs/2006.16908
5. Corrado & Hanna. Understanding when Dynamics-Invariant Data Augmentations Benefit Model-Free Reinforcement Learning Updates. https://arxiv.org/abs/2310.17786
7. Counterfactual Data Augmentation using Locally Factored Dynamics. Pitis et. al, NeurIPS 2020. https://arxiv.org/abs/2007.02863
6. MoCoDA: Model-based Counterfactual Data Augmentation. Pitis et. al, NeurIPS 2022. https://arxiv.org/abs/2210.11287

**Questions:**

1. If the environment is designed such that every state is reachable from every state with a single action, why not treat this task like a bandit problem? Why use RL?
1. Can the authors walk me through the purpose of the theoretical arguments at an intuitive level?

---

> ### Author Response · Authors · 2025-11-18
>
> We want to thank you for your extensive and thoughful comments and suggestions to help us improve
> our paper.
>
> First, we want to resolve a possible misunderstanding at the reviewers side: Not every state can be
> reached from every other state. In fact, the action space is limited in magnitude which prevents
> reaching arbitrary states in a single step. We discuss this fact only briefly in Section 2 and we
> thank the reviewer for bringing this up. As this may confuse also other readers, we are going extend
> our discussion in the revised version to make this clearer.
>
> Next, we want to start with an intuitive walkthrough which actually was part of our initial draft
> but which we had to remote due to space constraints. For a comparison between shortcuts and LIFT, we
> want to refer to our general response.
>
> ## Walkthrough of theoretical arguments behind shortcut augmentations
>
> In active positioning, good trajectories are these that reach the optimal position in as few steps as possible. However, most logging policies
> produce long and redundant trajectories, which however always contain states that are close to the
> goal. Our core idea is to train agents on synethetic trajectories destilled from these imperfect
> data which are more direct and goal-reaching. Very inuitiively, we want to agent to ``skip'' parts
> of the trajectory that do not add much value, like going straight instead of making zig-zag
> movements done in the logged data. This, however, is not straight-forwarod due to distortions in the
> dynamics and possible instability of the value function.
>
> For instance, assume a collected trajectory of a logging policy has a sub-trajectory $(s_i, W),
> (s_{i+1}, W),\ldots, (s_j, W)$ along with actions $a_i,\ldots,a_{j-1}$ which is a long zig-zag
> movement (or any other detour) from $s_i$ to $s_j$.
>
> Cleary, instead of doing the zig-zaging, directly moving from $s_i$ to $s_j$ would be better. However,
> naively applying the accumulated action $a'=a_i+a_{i+1}+\ldots +a_{j-1}$ at $s_i$ will likely not
> land us at $s_j$ due to distortions in the dynamics (it will though if $L_f=0$).  Already small
> misplacements, i.e. if we land close to $s_j$ but not exactly at $s_j$, can lead to significant
> value degradation if the value function is not stable in the vicinity of $s_j$.
> Even worse, applying $a'$ at $s_i$ may even lead us into the opposite
> direction, away from $s_j$ which no guarantee that the new state has a higher value than $s_i$.
> Here, the length of the action $a'$, the value gap between $s_i$ and $s_j$, the stability of the
> value function around $s_j$, and the distortion in the dynamics at $s_i$ all play a role.
>
> The main purpose of Section 3 is to identify situations (Definition 3.2 and Definition 3.3) when
> applying $a'$ at $s_i$ is beneficial, i.e., landing at a state where the value can be guaranteed to
> be larger than $s_i$ (Theorem 3.5 and Corollary 3.6).
> Our mathematical strategy is to get a bound on the misplacement error when applying $a'$ at $s_i$
> and to compare the value at the "landed" state with the value at $s_j$.
> Here, we we have to compare two mathematical objects: The
> misplacement from $s_i$ using $a'$ and the value function in the vicinity of $s_j$.
> Section 4 then shows how these insights can be fuled into a practical algorithm (Algorithm 1 -
> note that here we also check whether the accumulated action is contained in the action space).
>
> We are looking forward to hearing your feedback on this explanation. If its beneficial, we add a
> similar explanation at the beginning of Section 3 in the revised manuscript. We are more than happy
> to further clarify any remaining questions you may have that help you reconsidering your score.
>
> ## Related works
> We thank you for the valuable pointers to related works - its challenging to keep track of all
> recent developments and we really appreciate your help here. We will expand our related work section
> by discussing the mentioned references in the revised version of the paper. Specifically, we will
> discuss the differences of our method to GuDA and MoCoDA.
>
>
> ## Applicability beyond active positioning
>
> Regarding the applicability of LIFT beyond active positioning, we want to point to our general
> response where we discuss this question in detail.

---

### Author Response · Authors · 2025-11-18
**General Response (1/2)**

We would like to thank all reviewers for their thoughtful comments and suggestions to improve the
quality of our paper. We are excited that all reviewers appreciate that our augmentations
make offline RL practically useful for active positioning and that our idea and approach is
interesting and promising.
We will respond to each reviewer separately in the next days, but before we would like to
address some general comments that were raised by multiple reviewers.



## Area of applicability

We first want to highlight that active positioning as defined in Section 2 constitute is very broad
and practically important class of problems. It encompasses many core applications in robotics and
automation, including robotic manipulation (e.g., placing or inserting objects), autonomous
navigation to target poses, and industrial assembly and manufacturing, where precise placement is
central. Whenever an agent must bring an object or its end-effector to a desired configuration, an
active positioning problem arises. We therefore view our contribution as relevant to a wide range of
real-world control problems, rather than a single niche task.
We appreciate the reviewers' comments that LIFT may appear tailored specifically to active
positioning tasks. While our experiments indeed focus on this setting, we would like to clarify why
we believe this focus is still of broad interest and which aspects of our ideas are more generally
applicable beyond the specific benchmarks we study. For that, we want to first clarify the two
different augmentations types - *LIFT* in general and *shortcuts* in particular - and elaborate a bit
more about their generality. We apologize that these aspects were not sufficiently covered in the
original submission and we will make sure to address these points more clearly in the revised
version of the paper.

---

> ### Author Response · Authors · 2025-11-18
> **General Response (2/2)**
>
> ### Shortcuts
>
> Shortcut augmentations (Algorithm 1) are designed to capture the core structure of active
> positioning problems by exploiting literally *shortcuts* in long and inefficient trajectories
> generated by suboptimal logging policies. We refer to our response to reviewer `j18R` for an
> intuitive walkthrough of the general mechanism behind shortcut augmentations, which is the central
> part of our theoretical analysis in Section 3.
> Although shortcut augmentations are somewhat tied to the structure of active positioning problems as defined
> in Section 2 - which is a very broad class of problems though - we provide
> profound theoretical and empirical evidence that these augmentations significantly improve both the
> performance and data-efficiency of offline RL algorithms for this important class of problems.
> Clearly, our research is inspired by very concrete practical positioning problems, specifically those
> that arise in active alignments of optical components, like cameras or lasers. A particular
> difference to other robotic and locomotive tasks is the high precision that is required when
> positioning objects under varying contexts, putting more emphasis on precise relative positioning of the
> end-effector instead of rough absolute positions in a global coordinate frame.
> To come to an accommodation with the reviewers, we selected two robotic
> environments that inhibit somewhat these core assumptions and decided to provide a proof of
> principle for their effectiveness in the revised version of the paper (see below).
>
>
> ### LIFT
>
> The idea behind LIFT (Algorithm 2) is to enrich static logging policies with exploratory steps
> coming from an augmentor, i.e. a policy trained on the logged data in parallel. This builds
> upon the ideas from iterative reinforcement learning [1].
> Although this approach seems appealing, we believe that there is a central part that has been
> underexplored in current literature, namely that static logging policies may not deal well with
> intermediate exploratory steps. This is particularly true in the context of active positioning
> tasks. Instead of doing random exploratory steps, we want to insert steps that ease the
> hand-over between the logging policy and the augmentor and improve the quality of the
> collected data (Figure 5b shows that LIFT holds this promise).
> In order that the augmentor provides useful steps, it has to be trained well already with limited
> data. Our idea is to show the augmentor data of *good behavior* by applying augmentation to the
> logged data that emphasizes such behavior. In the concrete context of active positioning, we found
> that shortcut augmentations (Algorithm 1) are a very effective way to achieve this. However, we want
> to stress that LIFT is not tied to this form of backbone-augmentations, making it a modularized
> framework that can be adapted to a variety of tasks.
> Our current empirical benchmarks for LIFT are driven by active positionings and we strongly believe
> that this idea has huge potential beyond this setting. Nevertheless, we agree with the reviewers
> that currently, we do not provide empirical evidence for this claim. We hope the reviewers
> understand that the evaluation of LIFT for robotic tasks requires more careful design and
> engineering efforts and thus we have to leave this for future work.
> [1] Zhang et al., Active learning for iterative offline reinforcement learning, NeurIPS 2023.
>
>
> ## Extended experiments
>
> Multiple reviewers wished to see the performance of our approaches on additional environments and compared
> to more baselines. We are currently working on extending our experiments in these directions and
> will report the results as direct responses to the individual reviewers once finished.
> So far, we have decided to focus on the following extensions:
>
> - Evaluations on a stochastic environment (essentially we will use our PO-environment adding
>   stoachastic transitions) as suggested by reviewer `j18R`.
> - Comparisons with diffusion-based policies as suggested by reviewer `9oTn`.
> - Comparisons with diffsuion-based data augmentation methods, particularly Gta, as suggested by
>   reviewer `9oTn`.
> - Evaluation of shortcut augmentations for *FetchReach-v3* and *FetchPush-v3* from the OpenAI Gym robotics suite
>
> We appreciate feedback from the reviewers on these plans and are open to concrete suggestions for additional experiments.

---

### Author Response · Authors · 2025-12-03
**Summary of the Revisions**

We again want to thank all reviewers for their constructive comments and suggestions.
Overall, all reveiewers appreciated the novelty and interestingness of our approach and highlighed
that we not solely relied on empirical evaluations but also provided deep theoretical insight.


However, some weeknesses in the initial presentation of our paper hindered a clear view on the scope
of our work. Particularly, the reviewers main concerns have been:

- A): Some reviewers where worried that the scope of our experiments is too narrow and asked for
  extended evaluations.
- B): Some weeknesses in the initial presentation of our paper hindered a clear view on the scope
of our work and made it difficult to follow.
- C): Some reviewers asked for extended comparisons to existing methods, particularly
  diffusion-based techniques.

The recent impairments of OpenReview prevent to continue our started discussions with the reviewers
and we did not have a chance to get direct feedback regarding our intended changes. This is
particularly unfortunate, as some reviewers (`j18R`) expressed interest in continuing the discussion
and we believe that a discussion could helped resolving possible misunderstandings at the reviewer
sides.

Below, we summarize the main changes we made to our manuscript in order to address the reviewers'
concerns. We strongly belive that these changes significantly improve the quality of our paper and
would have lead to a more positive evaluation from the reviewers.

## A: Robotic experiments

Reviewer `j18R` explicitely asked for at least one additional experiments in a less structured environments which
would strengthen our paper’s empirical claims. Also, reviewer `9oTn` is of the opinion that
our initial experimental setup does not sufficiently demonstrate the generalizability of our approach.

Thus, our paper now also includes additional experiments with the robotic Fetch environment to
provide a proof of princple of the applicability of our method beyond the active positioning
settings we considered originally. Here, we consider image-based observations and scenarios where one and
two target objects need to be reached.

## B: Improved clarity

Some reviewers pointed out that the distinction between LIFT and shortcuts as well as the need for
the theoretic analysis was not properly laid out in the initial version of our manuscript. Thus, we revised all sections in order to
improve clarity. Specifically, we added additional visualisations (Figure 1 and Figure 2) to better understand our
setup and the basic idea behind LIFT. Moreover, we added the walkthrough example requested by reviewer `j18R`
at the beginning of Section 3 and extended the discussion on the similarties and
differences of LIFT and shortcuts.

## C: Extend comparisons to existing methods

As suggested by reviewer `9oTn`, we now also compare our method to diffusion-based techniques, both
as policies and as data augmentation, and show that our method also stays highly competitive in
these scenarios.

Some reviewers also pointed us to prior work that is related to our method. Consequently, we
extended and restructured our related work section appropriately.

---

### Meta-Review · Area_Chair_K93F · 2026-01-07

**Summary:**

**Summary of the paper**

This paper proposes LIFT, a framework for improving offline reinforcement learning via trajectory augmentations termed shortcuts. LIFT introduces (i) a trajectory-level augmentation that replaces suboptimal subsequences with shorter, higher-value transitions, and (ii) a policy-time augmentation that intermittently injects high-value actions during data collection. The authors provide theoretical conditions under which shortcut augmentations improve value and present empirical results primarily on active positioning tasks, reporting gains over standard offline RL baselines. Variants such as LIFT-SC are evaluated with multiple ablations on augmentation frequency and sampling schemes.

**Summary of the reviewers' concern**

Reviewer j18R mentioned that the empirical evaluation is narrowly focused on a specialized class of active positioning problems with structured rewards and dynamics, making it unclear whether LIFT generalizes to broader offline RL settings. The reviewer also found the theoretical section difficult to follow and insufficiently motivated, noting that key definitions and propositions appear to formalize intuitive notions (e.g., shortcuts corresponding to positive-advantage actions) without clearly explaining why this machinery is necessary. Additionally, this reviewer highlighted significant gaps in related work, particularly prior literature on data augmentation and coverage in offline RL, and questioned the fundamental limits imposed by the logging policy.
Reviewer Paah pointed out that the presentation and clarity are major weaknesses. This reviewer found the introduction and experimental setup confusing, the figures hard to read, and the role of the theoretical analysis insufficiently motivated at an intuitive level. This reviewer questioned the necessity of theoretical analysis. The empirical section was described as not very convincing, largely due to limited explanation of environments and setups.
Reviewer 9oTn expressed concerns about scope, generality, and comparative evaluation, recommending rejection. This reviewer argued that LIFT appears to rely heavily on domain-specific heuristics tailored to active positioning, limiting its relevance to the broader ICLR audience. They also emphasized missing baselines from the extensive literature on offline RL data augmentation (including noise-based and generative approaches) and questioned why more recent offline RL methods (e.g., diffusion-based policies) were not evaluated.

**Reviewer Concerns:**

The rebuttals partially address clarity and positioning issues, but core concerns remain unresolved. In response to Reviewer j18R, the authors clarify a misunderstanding about reachability (states are not all reachable in one step) and provide an intuitive walkthrough of the theoretical arguments, explaining when and why shortcut application can be beneficial. However, the rebuttal does not fundamentally resolve the concern that the theory largely formalizes intuitive, local structure, nor does it convincingly demonstrate general applicability beyond active positioning. In response to Reviewer Paah, the authors acknowledge presentation issues and promise clearer exposition, additional experimental details, and improved figures. While helpful, these are largely editing fixes and do not substantially strengthen the empirical evidence. In response to Reviewer 9oTn, the authors report additional experiments with diffusion-based methods. However, the other concerns remain.

**Reviewer Scores:**

The reviewers are unlikely to change the score (see my comment above).

---

### Decision · Program_Chairs · 2026-01-26

Reject